# Assessing Functional Ankle Instability in Sport: A Critical Review and Bibliometric Analysis

**DOI:** 10.3390/healthcare12171733

**Published:** 2024-08-30

**Authors:** Élio Alexandre, Diogo Monteiro, Ricardo SottoMayor, Miguel Jacinto, Fernanda M. Silva, Luis Cid, Pedro Duarte-Mendes

**Affiliations:** 1ESECS—Polytechnic of Leiria, 2411-901 Leiria, Portugal; elioalexandre.19@gmail.com (É.A.); diogo.monteiro@ipleiria.pt (D.M.); ricardosottomayor88@gmail.com (R.S.); miguel.s.jacinto@ipleiria.pt (M.J.); 2Research Center in Sport, Health, and Human Development (CIDESD), 5001-801 Vila Real, Portugal; luiscid@esdrm.ipsantarem.pt; 3Research Unit for Sport and Physical Activity (CIDAF), Faculty of Sport Sciences and Physical Education, University of Coimbra, 3040-248 Coimbra, Portugal; geral.fernandasilva@gmail.com; 4Sport Sciences School of Rio Maior, Polytechnic of Santarém (ESDRM-IPSantarém), 2001-904 Santarém, Portugal; 5Department of Sports and Well-Being, Polytechnic Institute of Castelo Branco, 6000-266 Castelo Branco, Portugal; 6Sport Physical Activity and Health Research & Innovation Center, SPRINT, 2040-413 Santarém, Portugal

**Keywords:** ankle, functional instability, questionnaire

## Abstract

Functional Ankle Instability (FAI) is the subject of extensive research in sports and other environments. Given the importance of accurately measuring this latent construct, it is imperative to carry out a careful assessment of the available tools. In this context, the aim of this review was to take an in-depth look at the six most cited measurement tools to assess FAI, with a specific focus on patient-reported outcome measures related to ankle and foot. Four electronic databases (Web of Science, Scopus, Pubmed, and SportDiscus) were searched (up to November 2022) to identify the six most cited questionnaires for assessing FAI. Our analysis showed that the most cited questionnaires are the following: the Lower Extremity Functional Scale (LEFS), the Foot Function Index (FFI), the Foot and Ankle Ability Measure (FAAM), the Foot and Ankle Outcome Score (FAOS), the Olerud and Molander Ankle Score (OMAS), and the Cumberland Ankle Instability Tool (CAIT). Each questionnaire was thoroughly assessed and discussed in three sections: Development, Reliability, and Summaries. In addition, bibliometric data were calculated to analyze the relevance of each questionnaire. Despite variations in terms of validity and reliability, conceptualization, structure, and usefulness, the six questionnaires proved to be robust from a psychometric point of view, being widely supported in the literature. The bibliometric analyses suggested that the FAOS ranks first and the FFI ranks sixth in the weighted average of the impact factors of their original publications.

## 1. Introduction

Ankle instability is a condition characterized by a recurrent giving way of the outer (lateral) side of the ankle [1]. It can be considered functional, mechanical, or chronic and can increase the risk of injuries, limitations in the regular practice of physical activity, pain, mental impact, biomechanical alterations, and economic cost [2]. On the other hand, functional ankle instability (FAI) is defined as an impairment in proprioception, neuromuscular control, postural control, and strength [3,4]. Mechanical ankle instability is described as an excessive anterior movement of the talocrural joint (i.e., the range of movement of the joint goes beyond the physiological range of movement expected for that joint) and can be assessed using an instrumental stress test (arthrometry) or a manual stress test [2]. Chronic Ankle Instability (CAI) is a condition of mechanical and functional instability that results in frequent ankle sprains [2,5,6] for at least 1 year after the initial sprain [1,2]. CAI, which develops from sprains, is one of the most reported conditions in the world of sports [7,8], highlighting the importance of understanding its causes, particularly through a clinical assessment of the ankle joint [9,10,11,12,13]. Studies indicate that FAI is related to CAI, and this relationship is one of the main intrinsic risk factors for developing tibiotarsal sprains [6,8,14,15].

Lateral ankle strains are commonly followed by midfoot ligaments injury [16], calcaneocuboid joint impairment [17], as well as tibial and fibular nerve injury [18]. These conditions may contribute to alterations in the function of the ankle–foot complex. Therefore, evidence reinforces the importance of clinical examination of the multiple segments of the ankle–foot complex in patients with lateral ankle strain or CAI [19].

A previous study by Hertel and Corbett [1] proposed the Model of Chronic Ankle Instability, which underscores the current understanding of the causes of ankle instability, reinforcing the importance of ankle–foot complex examination. The proposed model focuses on different components, including primary tissue injury, pathomechanical impairments, sensory-perceptual impairments, and motor-behavioral impairments [1]. A lateral ankle sprain involves a stretch or disruption of the collagen fibers of the lateral ligaments (e.g., the anterior talofibular ligament (ATFL) and calcaneofibular ligament (CFL)) provoking structural tissue damage [1]. The inherent pathomechanics impairments comprise a set of structural abnormalities to the ankle joints and surrounding tissues, including pathologic laxity of adjacent joints, arthrokinematic restrictions in the particular joint motions (e.g., talocrural, subtalar, midtarsal, and tarsometatarsal joints), osteokinematic restrictions (limitations in foot and ankle motion in multiple planes), secondary tissue damage, and tissue adaptations in the ankle and foot [1]. The sensory-perceptual impairments are related to patients’ senses or feelings about themselves, their bodies, and the injury, and include factors such as diminished somatosensation, pain, perceived instability, kinesiophobia, and reduced self-reported function [1]. Motor-behavioral impairments include altered reflexes, neuromuscular inhibition, muscle weakness, balance deficits, altered movement patterns in a spectrum of functional activities, and reduced physical activity [1]. All these components contribute to ankle dysfunction and CAI.

In recent years, various measures have been developed to assess ankle and foot function, together or separately. In the same sense, the analysis of quality and purpose is fundamental for their correct and adjusted application [20]. A systematic review encompassing studies published between 2012 and 2017 found more than 75 assessment measures for the ankle, pointing to the American Orthopedic Foot and Ankle Score (AOFAS) as the most used scoring system [21]. Currently, we come across several self-report questionnaires constructed to identify individuals with FAI [22]. One of the most cited in the academic world is the Foot and Ankle Ability Measure (FAAM) [23]. This questionnaire consists of a 21-item activities of daily living (ADL) scale and an 8-item sports scale, where patients rate their difficulty in performing ADL and specific sports activities with the AI. Another of the most cited measures is the Ankle Instability Instrument (AII) [24], a 12-item questionnaire divided into 3 parts (severity of initial ankle sprain, history of AI and instability during ADL), with information reported by patients and which doctors consider to be signs and symptoms present in someone with FAI. Methodologically rigorous measurement is required to assess, understand, and predict FAI predisposition [25,26].

Thus, critical evaluation of the strengths and weaknesses of different measurement approaches is essential for understanding functional instability. This analysis could increase the knowledge of researchers and professionals on the subject as there is a wide variety of measures used to assess functional instability, which can confuse researchers and professionals in their scientific and professional activities [27].

Therefore, the purpose of this study is to identify, overview, and evaluate the six self-report measures most frequently used to assess FAI, with a specific focus on patient-self-reported outcome measures related to ankle and foot.

## 2. Materials and Methods

### 2.1. Search Strategy

Through the application of the proposed bibliometric methods by Clancy et al. [28], we conducted a comprehensive search of the relevant literature in the field, systematically analyzing the number of citations and annual citations, as well as identifying the main sources of publication, journals, and relevant articles in the context of the scientific literature. Four electronic databases were searched until November 2022 (i.e., Web of Science, Scopus, Pubmed, and SportDiscus), considering the maximum period of retreat, in order to identify the most cited questionnaires to identify FAI. The search strategy included both MeSH and search index descriptors. By including the boolean indicators “AND” or “OR”, the search strategy resulted in the following: ((ankle OR “ankle joint” OR “ankle injury” OR “functional instability”) AND (sport* or exercise* OR training) AND (survey OR questionnaire* OR instrument* OR measur* OR scale OR “subjective evaluation” OR assess* OR tool OR “self-report*”) AND (“validation studies” OR psychometr* OR valid* OR “internal consistency”)). The reference lists of the articles obtained were also analyzed manually to find potential studies to include in this review.

### 2.2. Eligibility Criteria

This study considered studies that met the following inclusion criteria to be eligible for analysis: (i) Studies with any design (cohort study, cross-sectional study, case–control study, clinical trials); (ii) With any population, regardless of race, ethnicity, gender, or age group; (iii) Studies with any number of participants; (iv) No restriction regarding the country or year of publication; (v) Studies that used a tool to assess FAI; (vi) Studies with patient-self-reported outcome measures related to ankle and foot measures. On the other hand, studies with the following characteristics were excluded: (i) Studies not written in English; (ii) Articles with participants with other pathologies; (iii) Articles without a description of the assessment tool.

### 2.3. Data Extraction

Two authors carried out the search, selection of the articles, and data extraction independently, and the results were later compared and discussed. The authors extracted the following information from the primary studies: name of the measure, first author, year of publication, country first author, journal where it was published, total number of citations of the journal, information about the scale including test–retest reliability, items, subscales, answers/items, classification, number of citations, citations/years, impact factor of original journal, quartile, number of journals that cited it, number of manuscripts that cited the measures, and impact factor weighted average. EndNote v21.1 software was used for peer review, elimination of duplicate articles, and data extraction.

## 3. Results

An exhaustive search of the databases resulted in a total of 2319 studies. After the initial selection phase, which involved reviewing titles and abstracts, 53 studies were identified as potentially relevant to this work. Subsequently, applying the pre-defined eligibility criteria and carrying out a review of the full-text articles, the six most cited measures for assessing FAI were found. These studies were, therefore, included in this critical review for detailed analysis (Figure 1).

Figure 2 presents a comprehensive analysis of publication trends for the period 1983 to 2022, considering our search terms. In general, publications have gradually increased over the years.

The six most cited LFS evaluation questionnaires were selected for analysis using the Cited Reference Search in Web of Science and are summarized in Table 1: FAAM [23], Lower Extremity Functional Scale (LEFS; [29]), Foot Function Index (FFI; [30]), Foot and Ankle Outcome Score (FAOS; [31]), Olerud and Molander Ankle Score (OMAS; [32]), and Cumberland Ankle Instability Tool (CAIT; [33]).

The first study found that answered our study objective was published in 1984 [32], in the 20th century. In that same century, there were two more publications [29,30], while the remaining three were published in the 21st century [23,31,33]. The last publication that answered our study objective was published in 2006 [33]. Three of the six studies come from the USA, the American continent, which is the country and continent with the most studies. Two studies came from Sweden, on the European continent. Another study came from Australia, the continent of Australia.

Table 2 presents an overview of the six most cited evaluation measures, considering the results of test–retest reliability, number of items in the measure, subscales, answers/items (Likert scales), and classification.

To evaluate each instrument, other searches were carried out using the name of the questionnaire combined with terms related to the evaluation of the test, i.e., reliability, psychometrics and factor analysis. Table 3 presents the bibliometric data obtained from the Cited Reference Search of the Web of Science. For analysis, we took each original publication’s total number of citations as well as the subset with an impact factor. Different sources without impact factor (i.e., including abstracts from scientific events and book series) were excluded from the calculations. The following formula was used to calculate the weighted average impact factor for the original publication of each questionnaire: (i) multiply the number of articles (citations) in each journal by the journal’s impact factor in 2021; (ii) add up this value for all journals together and divide by the total number of articles (citations). This calculation resulted in a single number, allowing us to describe the impact of each questionnaire. Similar to Clancy et al. [28] study, no selection was made by journal or discipline.

Thus, the analysis of each measure was conducted in order from the highest to the lowest number of citations since the date of publication of the original study. The bibliometric data, presented in Table 3, indicates that LEFS is the most cited questionnaire under analysis (43 citations per year). However, the weighted average impact factor of the journals responsible for these citations is 3.59. Other questionnaires, such as FAOS and OMAS, presented highest weighted average impact factors (4.70 and 4.42, respectively), but many of the LEFS citations are in journals outside sports science (e.g., BMC Musculoskeletal Disorders, Disability and Rehabilitation or Physical Therapy). Thus, FAOS could be interpreted as the questionnaire with the greatest impact.

In the current review, each measure is evaluated in three domains: (i) description of the questionnaires, with basic information, structure, updated versions, scoring, among others; (ii) brief summary of reliability; and (iii) a synopsis of all the information is presented in summary. To assess internal consistency (IC), we reported the Cronbach’s alpha values (α), with 0.70 being the acceptable cut-off point for research purposes [34]. The indices of temporal stability and model fit are also reported whenever available. Test–retest correlation and intraclass correlation coefficients (ICCs) are interpreted as proposed by Vincent and Weir [35]: high (>0.90), moderate (0.80–0.90), or insufficient (0.80)

### 3.1. Lower Extremity Functional Scale

#### 3.1.1. Development

The LEFS is a specific measure applicable to a wide variety of orthopedic conditions of the lower limbs, including patients with different levels of disability [36]. Binkley et al. [29], created the items for the instrument through a process of reviewing previous questionnaires, as well as surveys of doctors and patients, where the World Health Organization’s model of disability (1948) served as the basis. The final version of the LEFS consists of 20 items, each scored from 0 to 4, giving a possible total of 80, indicating a high functional level [29]. In response to criticism directed at the LEFS due to its ease of translation, but with difficulties in properly choosing the metric equivalent for distance traveled, as well as disordered thresholds and response dependence on the number of items [37,38], Bravini et al. [39] developed the 15-Item LEFS Version. This new version consists of 15 items, showing good Infit and Outfit MnSq values through Rasch analysis. In the same sense, Repo et al. [40] concluded that the 15-item LEFS is a meaningful and feasible psychometric assessment for measuring foot and ankle function. Based on the LEFS-Ar [38], the 15-item LEFS-Ar was also created [41], showing excellent CI. Evidence was later shown to support its confidence interval (CI), test–retest reliability (CTR), and construct validity as a measure of lower limb function [42]. Although the 15-item LEFS shows validity for lower extremity disorders, Saarinen et al. [43] showed suboptimal longitudinal validity in foot and ankle orthopedic patients undergoing surgical treatment. In a systematic review with 27 articles included, 18 of them with high methodological quality, Mehta et al. [44] present results that support the reliability, validity, and responsiveness of the LEFS score to assess the functionality of musculoskeletal conditions of the lower extremities. The subscale scores (mean, followed by standard deviation in brackets) for the LEFS are provided in Table 4.

#### 3.1.2. Reliability

Acceptable CI was found in most of the studies used by LEFS [44]. Binkley et al. [29] reported CI values of 0.96, and CTR estimates showed a correlation coefficient of R = 0.86 (95% lower limit CI = 0.80). Lin et al. [45] reported CI values of 0.90 for people with ankle fractures. In the same sense, Yeung et al. [46] confirmed acceptable validity for inpatients undergoing orthopedic rehabilitation, with an CI of 0.88 (95% CI: 0.74, 0.95) and a standard error of measurement of 3.5 points on the LEFS (95% CI: 2.7, 4.9 points on the LEFS), with the error associated with a LEFS score observed at a 90% confidence level. For Italian people with ligament injuries, α values between 0.91 and 0.96 have been reported [37]. The Brazilian version of the LEFS [47] showed a CI of α = 0.96, an intraobserver reliability of 0.96, and an interobserver reliability of 0.98. Coefficient α values above 0.70 were obtained for Iranian outpatients with ankle and foot disorders [48]. Cruz-Díaz et al. [49] also confirmed high CI (α = 0.989) and CTR (CCI = 0.998, 95% CI: 0.996–0.999) in ankle and foot disorders, in a Spanish version. The CI, together with the CTR, proved to be satisfactory for Chinese patients undergoing ankle orthopedic rehabilitation (α = 0.98; CCI = 0.97) [50]. In patients with plantar fasciitis and heel pain, excellent levels of CTR were found (ICC = 0.96), with a standard error of measurement of 3.5 points and a minimum detectable change (MMD) of 9.8 points. The Arabic version also showed an excellent CI (0.95) [38]. The CTR was high (ICC = 0.93, 95% CI: 0.91–0.95) for Finnish patients with surgically treated ankle and foot pathology, with a standard error of measurement of 4.1 points [51]. Zhang et al. [52] present a high CI value (0.97) and CTR (CCI = 0.97) for non-specific sprains.

**Table 4 healthcare-12-01733-t004:** Sample composition and subscale scores for a variety of articles using the LEFS.

Reference	Sample	Total Score
Binkley et al. [29]	107 Participants 12 Physiotherapy clinics	39 (18.0)
Cacchio et al. [37]	250 Participants Physical Medicine Department and Rehabilitation ‘‘La Sapienza’’	Pre-treatment score 32 (20) Post-treatment score 63 (18)
Alnahdi et al. [38]	116 Participants Physiotherapy Department of King Khalid University Hospital and Prince Faisal Sport Medicine Hospital in Riyadh	33.04 (17.7)
Lin et al. [45]	306 Participants Physiotherapy Centers	30.68 (12.37)
Negahban et al. [48]	304 Participants Physiotherapy Clinics in Teerã e Ahvaz	46.9 (16)
Crúz-Diaz et al. [49]	132 participants Physiotherapy Clinics and Rehabilitation	56.2 (13.5)
Hou et al. [50]	159 Participants Orthopedic wards and rehabilitation outpatient clinics in university hospitals	31.6 (27.4)
Repo et al. [51]	166 participants Operated at the authors’ institutions	66.2 (15.4)
Zhang et al. [52]	213 Participants Cirurgical Department of Changzheng Hospital	45.1 (19.2)

Note. LEFS—Lower Extremity Functional Scale.

#### 3.1.3. Summary

The original version of the LEFS consists of 20 items, with a maximum score of 4 points for each item, and the higher the score, the higher the functional level. Bravini et al. [39] developed a 15-item version, called the LEFS 15-Item Version, which underwent Rasch analysis and showed good fit values. Other studies, such as that by Repo et al. [40], have also confirmed the psychometric validity and relevance of the 15-item LEFS for measuring foot and ankle function. The reliability of the LEFS has been widely examined in several studies, with CI values reported above 0.70, in different populations and languages, such as Italian, Spanish, Chinese, and Arabic. In addition, the CTR has shown consistent ICC and standard errors of measurement in various samples.

### 3.2. Foot and Ankle Ability Measure

#### 3.2.1. Development

The FAAM is a questionnaire developed to assess the functional capacity of the lower leg, ankle, and foot in individuals with injuries or conditions related to this region [23]. Its final version is made up of two subscales: the 21-item FAAM-ADL subscale and the 8-item FAAM-Sports subscale, where both are scored separately on a Likert scale from 0 to 4 points, with 0 being unable to do and 4 being without difficulty [23,53]. If all 21 items are answered, the maximum score is 84. After completing the questionnaire, the total score of the items is divided by the maximum possible score and multiplied by 100 to produce the ADL score, which ranges from 0 to 100 [23]. The Sport subscale is scored in a similar way, with the highest potential score being 32. After completing the questionnaire, the total rating of the items is divided by the highest potential score and multiplied by 100, with higher values representing a higher level of physical function for both subscales [23]. The Sport subscale is scored in a similar way, with the highest potential score being 32. After completing the questionnaire, the total score of the items is divided by the highest potential score and multiplied by 100, with higher values representing a higher level of physical function for both subscales [23]. With the aim of creating a shortened version of the FAAM, reducing response time and scoring in clinical settings, the Quick-FAMM was developed [54], a questionnaire reduced to 12 questions that includes five ADL items and seven sports-related items. Later, Hoch et al. [55] came up with a CI α = 0.94, demonstrating an acceptable ICC = 0.82 for the reduced 12-item version of the FAAM. A cross-sectional project [56] reported that the Quick-FAAM is a valid and reliable regional instrument for clinical use. Table 5 shows the scores for each subscale (mean followed by standard deviation in brackets).

#### 3.2.2. Reliability

When developing the FAAM, Martin et al. [23] found that the CTRs for the ADL and sports subscales were 0.89 and 0.87, respectively. In addition, they determined that the MMDs for the ADL and sports subscales were ± 5.7 and ± 12.3 points, with a 95% confidence interval. Nauck et al. [57], when investigating possible functional deficits related to CAI in German sports students, showed an excellent CTR, with ICC ranging from 0.590 to 0.998. Mazaheri et al. [58], to quantify the physical function of the foot in Iranian patients, obtained CI values of 0.97 and 0.94 for the ADL and Sports subscales, respectively. In addition, they found an ICC and standard error of measurement of 0.98 and 3.13 for ADL, and 0.98 and 3.53 for Sport. In French patients with degeneration, trauma, congenital deformities, regional syndromes, and tumors, Borloz et al. [59] reported α estimates of 0.97 for both subscales, with ICCs of 0.97 for ADL and 0.94 for sport. Uematsu et al. [60], when studying Japanese athletes in men’s and women’s basketball, men’s rugby, men’s soccer and men’s gymnastics, obtained an α for CI of 0.99 for ADL and 0.98 for sport, with a 95% confidence interval. Also, Çelik et al. [61], in Turkish patients with plantar fasciitis, Achilles tendinopathy, osteoarthritis, calcaneal spurs, hallux valgus, and diabetic foot, showed a CTR of 0.90 for both FAAM subscales, and α coefficients of 0.95 and 0.91 for the ADL and sport subscales, respectively. Moreira et al. [62] obtained ICCs of 0.88 and 0.82 in Brazilian patients, with α coefficients of 0.93 and 0.90 for the ADL and sports subscales, respectively. González-Sánchez et al. [64] also reported an ICC of 0.879 for ADL and 0.901 for sport, with a CTR ranging from 0.758 to 0.970 (ADL: 0.758–0.946; sport: 0.911–0.970) in Chinese patients with ankle diseases.

#### 3.2.3. Summary

The FAAM questionnaire consists of two subscales: FAAM-ADL and FAAM-Sports, and the answers are based on a 5-point Likert scale. Similarly, the FAAM has an adequate CTR, with ICCs ranging from 0.590 to 0.998. In addition, the α values were consistent, ranging from 0.91 to 0.99 for the ADL and sport subscales. The Quick-FAAM was also developed, a shortened version of the questionnaire, which demonstrated clinically acceptable reliability. A potential limitation of the FAAM is the lack of consideration of the basic functional level of individuals by not assigning data related to the minimum detectable change and the minimum clinically important difference (MCID) [23].

### 3.3. Foot Function Index

#### 3.3.1. Development

The FFI is a questionnaire that was initially developed to measure foot function [30] and later adapted to also cover the ankle joint [65]. The questionnaire consists of three subscales that measure the pathological condition in terms of pain, disability, and activity restriction, totaling 23 items [30]. The answers to each item are given using a visual analog Likert scale, ranging from 0 to 10, where 0 indicates “no interference” and 10 represents “maximum interference” [66]. The total score is calculated by adding up the scores for each subscale and then dividing by the number of questions, where a higher score refers to a greater functional limitation [65]. In 2006, the Foot Function Index-Revised (FFI-R) was developed by Budiman-Mak et al. [30], based on recommendations and restrictions related to the care of foot and ankle joint problems [67]. The FFI-R consists of 68 items based on a theoretical model, in which the assessment domains were clarified, and a new psychosocial scale (social/emotional) was added, showing potential to be sensitive to samples with a significantly higher severity of foot problems [68,69]. A multifaceted Rasch analysis by Ryu et al. [70] demonstrated solid psychometric properties of a 32-item short-form version of the FFI-R focused on foot function. In the same vein, a reliable and valid 17-item version of the FFI, developed by Venditto et al. [71], has been published to assess musculoskeletal conditions and disorders of the foot and ankle more briefly. Table 6 shows means and standard deviations related to the use of the questionnaire.

#### 3.3.2. Reliability

When developing their original version, Budiman-Mak et al. [30] reported CTR values of the FFI total score and subscales ranging from 0.87 to 0.69 and CI values ranging from 0.96 to 0.73 in patients with rheumatoid arthritis. In Chinese patients with ankle fractures, Wu et al. [72] found a high CI (0.94) and a satisfactory CTR (CCI = 0.82). Martinelli et al. [73] showed an α value of 0.95 for both FFI subscales, with good reproducibility, and ICCs of 0.94 and 0.91 for the pain and disability subscales, respectively, in Italian patients with ankle problems referred for surgery. A study of Danish patients with tibiotarsal sprains reported excellent CI in all three subscales (pain: 0.99, disability: 0.98, and activity limitation: 0.98), as well as an excellent CTR (pain subscale: ICC 0. 98 [95% confidence interval: 0.97–0.99]; activity limitation subscale: ICC: 0.95 [95% CI: 0.91–0.98]; disability subscale: ICC 0.97 [95% CI: 0.95–0.98]) [74]. Martinez et al. [75], in Brazilian patients with various foot and ankle problems for more than 6 months, reported excellent intra- and inter-rater correlations, with an ICC of 0.99 to 0.97, and a score reliability considered highly satisfactory, with an α range of 0.80 to 0.61. In patients with functional limitations in the ankle, Gonzalez-Sanchez et al. [64] showed CI ranging from 0.996 to 0.998, test–retest analysis from 0.985 to 0.994 and an MMD of 2.270, with a standard error of measurement of 0.973. Finally, the Turkish version of the FFI, validated by Külünkoğlu et al. [76], showed an α value ranging from 0.821 to 0.938, with satisfactory reproducibility and ICC values between 0.960 and 0.985.

#### 3.3.3. Summary

FFI consists of three subscales that measure the pathological condition in terms of pain, disability, and activity restriction, totaling 23 items. The answers to each item are given on a Likert scale, ranging from 0 to 10, with a higher score indicating greater functional limitation. The contribution of the activity limitation subscale to measuring foot function is unclear [30]. Subsequently, the FFI-R was developed, a 68-item version based on a more comprehensive theoretical model, including a psychosocial scale. In addition, there is a shorter version with 32 items focused on foot function and a reliable and valid version with 17 items to assess musculoskeletal conditions of the foot and ankle. Several studies have reported high reliability of the FFI, including satisfactory CI and CTR, in different populations and clinical conditions.

### 3.4. Foot and Ankle Outcome Score

#### 3.4.1. Development

The FAOS is an adaptation of the Knee Injury and Osteoarthritis Outcome Score (KOOS) [77], developed for application to patients with ankle and foot problems, formulated for the first time using a sample of 213 patients undergoing anatomical reconstruction of the lateral ankle ligaments [31]. The questionnaire comprises 42 items that assess relevant outcomes in five distinct sub-scales, covering pain, additional symptoms (edema, stiffness, instability), ADL, sports and recreational activities, and foot- and ankle-related quality of life (QoL) [31]. Patients are asked to rate their symptoms on a 5-point scale: none (0), mild (1), moderate (2), severe (3), or extreme (4), and each of the five subscale scores is calculated as the sum of the items included; the raw scores are transformed into a scale from 0 (no symptoms) to 100 (extreme symptoms) [31,78]. Studies show that the FAOS is widely used to assess conditions in the ankle and foot [79,80,81], and values for assessing MMD are considered clinically positive [82,83]. The subscale scores (mean and standard deviation) are shown in Table 7.

#### 3.4.2. Reliability

Roos et al. [31] demonstrated the high reliability of the five subscales of the FAOS questionnaire, with α values of 0.94, 0.88, 0.97, 0.94, and 0.92 for pain, symptoms, ADL, sporting and recreational function, and QoL related to the foot and ankle, respectively. In the test–retest, Spearman’s correlation coefficients for the scores obtained in the first and second administration were 0.96, 0.89, 0.85, 0.92, and 0.92 for the subscales of pain, symptoms, ADL, sports and recreational function, and the foot- and ankle-related QoL, respectively [31]. The ICCs for the same subscales were 0.78, 0.86, 0.70, 0.85, and 0.92 [31]. In Turkish patients with ankle dysfunction persisting for at least one month, the FAOS proved to be a valid and reliable instrument, showing a random ICC for the five subscales ranging from 0.70 to 0.96, with the α coefficient ranging from 0.79 to 0.97 [84]. In Portuguese patients with lateral ankle ligament injury due to inversion sprain, the questionnaire showed good reproducibility and reliability for all the subscales, both intra-interviewer and inter-interviewer (*p* < 0.05) [85]. The test–retest results show high reliability for all FAOS subscales, with ICCs ranging from 0.92 to 0.96, and the minimum α value of 0.70 was exceeded by most subscales for a Persian version in ankle disorders of musculoskeletal origin [86]. In Korean patients with joint instability, all subscales except the QoL subscale (α = 0.615) showed satisfactory CI (α > 0.7) [87]. To evaluate the results in acquired flexible flatfoot deformity in American adults, content validity, reliability, and acceptable responsiveness were presented, with ICC values ranging from 0.79 (QoL, symptoms) to 0.88 (sport/recreation) [78]. The Dutch version of the FAOS is also a reliable and valid questionnaire for assessing symptoms and functional limitations of the foot and ankle, with α ranging from 0.90 to 0.96 and ICC ranging from 0.90 to 0.96 [88]. Godlightly et al. [89] report sufficient reliability and validity, with a large sample based on an American community aged ≥ 50 years (α ranging from 0.89 to 0.92 and ICC ranging from 0.63 to 0.81). In a more recent study, with a larger sample of Dutch subjects, the FAOS subscales were found to have acceptable measurement properties, with ICC ranging from 0.83 to 0.88 and α 0.76 [82]. Van Bergen et al. [90] demonstrated the CTR and CI for each subscale, with ICC of 0.88–0.95 and α of 0.94–0.98, in German patients with ankle dysfunction, including instability. The minimum detectable changes for each subscale were 17.1 to 20.8 at the individual level and 2 to 2.4 at the group level. Mani et al. [91] present α values ranging from 0.76 (Symptoms) to 0.95 (ADLs) for American subjects with ankle osteoarthritis. For the assessment of foot and ankle arthritis, the FAOS demonstrated sufficient levels of content and construct validity; however, reliability was satisfactory only for the ADL subscale, being insufficient for the secondary subscales (ICC = 0.33 and α for the pain subscale = 0.94; symptoms = 0.58; ADL = 0.96; sport and recreation = 0.79; and QoL = 0.93) [92]. For ankle fractures, the questionnaire has an ICC of 0.88 (95% confidence interval: 0.79–0.93) for the pain subscale, 0.95 (95% CI: 0.91–0.97) for symptoms, 0.95 (95% CI: 0.90–0.97) for ADL, 0.95 (95% CI: 0.90–0.97) for sport, and 0.94 (95% CI: 0.90–0.97) for QoL in Danish patients [93]. Pellegrini et al. [94] show high reliability capable of assessing different foot and ankle conditions in Chilean patients, with an α = 0.98. For Spanish podiatric patients, it is considered a strong and valid questionnaire with adequate repeatability (α ranging from 0.92 to 0.95 and ICC ranging from 0.78 to 0.92 [95]. In a more recent study, with a larger sample, a high CTR of the FAOS was observed in patients with ankle fractures (ICC values ranging from 0.71 to 0.78) and high α, with values ranging from 0.8 to 0.88 [96].

**Table 7 healthcare-12-01733-t007:** Sample composition and subscale scores for a variety of articles using the FAOS.

Reference	Sample	Pain	Symptoms	ADL	Sport	QoL
Mani et al. [78]	126 Participants with flat feet	67.68 (18.72)	65.68 (19.01)	78.28 (17.28)	45.40 (29.70)	34.21 (21.16)
Sierevelt et al. [82]	110 Participants with complaints in ankle and hindfoot	58.1 (19.9)	54.1 (21.1)	71.1 (19.8)	37.4 (23.2)	30 (19.2)
Karatepe et al. [84]	55 Participants with foot and ankle disfunctions	58.4 (21.8)	66.9 21.8)	64.0 (23.4)	56.4 (26.7)	50.9 (23.0)
Imoto et al. [85]	50 Participants with ankle sprains	83.42 (20.17)	82.68 (18)	90.66 (19.38)	71.60 (29.28)	61.64 (28.65)
Negahban et al. [86]	93 Participants with foot and ankle disorders	62.2 (16.68)	55.93 (13.27)	68.42 (22.11)	30.23 (20.96)	31.58 (18.35)
Lee et al. [87]	294 Participants with foot and ankle disfunctions	65.4 (19.5)	74.2 (17.3)	79.9 (18.9)	61.5 (26.3)	41.6 (27)
Van Den Akker-Scheek et al. [88]	89 Participants undergoing ankle or hallux valgus arthroscopy	67.7 (24)	69.8 (22.9)	77.2 (22.6)	55.9 (34)	51.1 (28.9)
Golightly et al. [89]	1670 Participants with foot and ankle disorders	86 (20)	87 (16)	95 (10)	74 (34)	83 (23)
Van Bergen et al. [90]	150 Participants with various disorders in foot and ankle	57.6 (23.0)	60.2 (21.1)	69.4 (22.6)	47.1 (29.0)	38.5 (23.8)
Navarro-Flores et al. [95]	79 Participants disorders	55.90 (21.05)	55.65 (19.40)	65.43 (15.92)	57.59 (24.12)	46.67 (26.51)
Larsen et al. [96]	76 Participants ankle fracture	62.1 (27.2)	52.5 (19.3)	52.1 (25.6)	6.4 (12.0)	19.2 (17.7)

Note. FAOS—Foot and Ankle Outcome Score; ADL—Activities Daily Living; QoL—Quality of Life.

#### 3.4.3. Summary

FAOS questionnaire consists of 42 items covering five distinct subscales: pain, additional symptoms, ADL, sport and recreational activities, and QoL. Patients rate their symptoms on a 5-point scale, and the subscale scores are calculated as the sum of the items and transformed into a scale from 0 to 100. The FAOS is widely used to assess ankle and foot conditions, with values considered clinically positive for MMD. Several studies have shown high CI and CTR for the FAOS subscales in different populations and clinical conditions. For example, patients with ankle dysfunction had an ICC ranging from 0.70 to 0.96 and an α coefficient ranging from 0.79 to 0.97. In addition, studies in different countries, such as Korea, the Netherlands, the United States, Chile, and Spain, have also reported high reliability and validity in different clinical conditions. A limitation of the original FAOS is that its responsiveness has not been evaluated [31].

### 3.5. Olerud and Molander Ankle Score

#### 3.5.1. Development

The OMAS was originally developed as a patient-response scale for assessing symptoms after ankle fracture [32]. Subsequently, it was also used for assessment following acute injuries and ankle function [97,98]. The study that developed the OMAS involved a sample of 90 subjects with ankle fractures who answered 9 questions with different scores (pain (25 points), stiffness (10 points), swelling (10 points), climbing stairs (10 points), running (5 points), jumping (5 points), squatting (5 points), supports (10 points), and work level/ADL (20 points), with the aim of assessing joint function [32]. The final score is calculated by adding up the points for each item assessed, giving a result from 0 (totally impaired function) to 100 (excellent or completely unimpaired function). Score ranges have been established, considering values from 0 to 30 as poor, 31 to 60 as fair, 61 to 90 as good and 91 to 100 as excellent [32]. However, some studies have raised concerns regarding the lack of evidence demonstrating measurement properties of this instrument [99,100]. In contrast, McKeown et al. [101] demonstrate adequate convergent validity and satisfactory CI in their measure of ankle function. In addition, the minimally important change corresponds to 9.7 points [101]. Table 8 shows the means and standard deviations relating to the use of the questionnaire.

#### 3.5.2. Reliability

OMAS has been widely used in research and clinical studies related to ankle injuries, demonstrating positive results in assessing functionality and QoL after treatment of these injuries [97,98,102]. Van der Wees et al. [103] indicate that, for patients with acute ankle injuries, the OMAS has mean final scores of 91.1 (standard deviation: 12.5) and 15 (standard deviation: 27), respectively, with Pearson correlations ranging from 0.36 to 0.41. For Swedish patients 12 months post-ankle-fracture, the OMAS proved to be a reliable and valid measure, with a CTR correlation coefficient of 0.95, ICC of 0.94, and CI of 0.76, as observed by Nilsson et al. [104]. In Swedish subjects with ankle fractures up to two years post-surgery, Büker et al. [105] report a high CTR correlation (R = 0.882), a CCI of 0.942, and an α of 0.762. In a retrospective cohort study of 959 Norwegian patients undergoing surgery for unstable fractures, it was found that three years after surgery, the OMAS showed α and CTR correlations ranging from 0.82 to 0.96 and 0.91 to 0.93, respectively [106]. When evaluating the psychometric properties of the instrument, McKeown et al. [101] showed an α of 0.76 for ankle function in British patients recovering from fractures between 6 and 16 weeks. The Brazilian version of the OMAS showed an excellent CTR (ICC = 0.99), as demonstrated by Castilho et al. [107], highlighting its good applicability in assessing functional capacity after the treatment of ankle fractures. According to the Turkish version, in ankle fractures up to four years post-surgery, an excellent CI is observed with an α of 0.84 and CTR (CCI) of 0.98 ([108]).

**Table 8 healthcare-12-01733-t008:** Sample composition and subscale scores for a variety of articles using the OMAS.

Reference	Sample	Total Score
McKeonwn et al. [101]	620 Participants in tibiotarsal fracture rehabilitation	43 (26.36)
Van der Wees et al. [103]	107 Patients with acute injuries	55.51 (26.68)
Nilsson et al. [104]	42 Participants surgically treated for ankle fractures	75 (19)
Büker et al. [105]	91 Participants with maleolar fracture	72.58 (23.27)
Garrat et al. [106]	959 Patients with unstable tibiotarsal fractures	74.12 (24.91)
Castilho et al. [107]	40 Participants who received surgical treatment for a tibiotarsal fracture	83.1 (18)
Turhan et al. [108]	100 Participants with maleolar fracture	74.1 (23.7)

Note. OMAS—Olerud and Molander Ankle Score.

#### 3.5.3. Summary

Available evidence indicates that OMAS scores can be interpreted to assess functionality and QoL in patients with ankle fractures. Although it has been widely used in research and clinical studies, some concerns have been raised regarding its measurement properties. However, more recent studies have demonstrated adequate convergent validity and satisfactory CI in the measurement of ankle function. In addition, the OMAS has been shown to be reliable and valid in different populations, with consistent ICC and α. For patients with acute ankle injuries, the OMAS showed high average scores, indicating good functionality. A limitation that will be useful to investigate is the relationship between subjective and objective instability, as this relationship may influence the decision of patients to be advised not to return to sport too prematurely, as they may be at greater risk of recurrent injury [97].

### 3.6. Cumberland Ankle Instability Tool

#### 3.6.1. Development

The CAIT is a widely used measure for assessing FAI severity [33]. The construction of the questionnaire involved identifying questions used in previous studies on ankle injuries. Based on these sources and after carrying out pilot studies on patients with uninjured ankles, the final version of the CAIT was developed, consisting of 9 items where each item is scored individually, and the maximum possible score is 30. Lower scores indicate more severe functional instability of the ankle [33]. To analyze whether the original cut-off score (≤27) is suboptimal for use in the TI population, Wright et al. [109] present highly favorable clinometric properties in their study, indicating that clinicians using the CAIT should employ the recalibrated cut-off score to maximize the test’s characteristics. Mirshahi et al. [110] later corroborated these findings, demonstrating the same pattern for the athletic population involving various sports. A digital version of the CAIT revealed good to excellent psychometric properties [111], concluding that clinicians using the digital version can confidently use it to accurately assess patients. The subscale scores (mean followed by standard deviation) for the CAIT are provided in Table 9.

#### 3.6.2. Reliability

Acceptable CI values were found in most studies using the CAIT. Its original version demonstrated excellent construct validity, with CI 0.83 and CTR (ICC = 0.96) [33]. For ankle pain in Brazilian subjects, Noronha et al. [112] reported high CI (0.86 for right ankle and 0.88 for left ankle) and reliability (ICC = 0.95, 95%; CI 0.93–0.97). In Spanish patients with CAI, Cruz-Díaz et al. [113] found a high CI (0.766) and reliability (ICC = 0.979, 95%; confidence interval = 0.958–0.990). For Australian pediatric patients with Charcot–Marie–Tooth disease, who frequently suffer ankle sprains and have chronic instability, Mandarakas et al. [114] found good CTR (ICC = 0.73, 95% CI: 0.11–0.91). For Thai children (8 to 16 years old) playing school sports, Kadli et al. [115] found a good CI (0.767) and a substantial CTR with a CCI of 0.865 (95% CI = 0.809–0.904). Rodríguez-Fernández et al. [116] found excellent CI (0.8–0.84) and reliability (ICC = 0.95) in Spanish patients who practiced different sports and had a history of ankle injuries. For Korean athletes from various Olympic sports, Ko et al. [117] showed a high CI (0.89) and ICC (0.94). Haji-Maghsoudi et al. [118] assessed the reproducibility of the questionnaire in Iranian taekwondo athletes and found an α of 0.64 (close to the acceptable level of CI) and a test–retest ICC for CAIT scores of 0.95. Hadadi et al. [119], in a Persian version, determined functional instability and found an α above the cut-off point of 0.70 for both joints, along with a high ICC for the right ankle (0.95) and (0.91) for the left. In the Dutch version of the CAIT to assess ankle instability, Vuurberg et al. [120] found an excellent ICC of 0.94 and high CI (0.86). Korakakis et al. [121] reported excellent ICC (0.92) and reliability (0.75–0.98) in Arab athletes from various sports. Lin et al. [122] showed an excellent CTR (ICC = 0.91, 95%; confidence interval = 0.87–0.94, *p* < 0.001) and a good CI (0.87) for differentiating stable and unstable joints in Chinese athletes. In a similar study with a Greek population, Tsekoura et al. [123] found good discriminative validity, high CI (0.97) and excellent CTR (ICC = 0.97, 95% CI = 0.97–0.98). Mirshahi et al. [110] also observed good CI (0.78 right ankle and 0.79 for the left) and substantial reliability (ICC = 0.88, 95%; CI: 0.86–0.90) in Iranian volleyball, basketball, and track and field athletes. Geerinck et al. [124] demonstrated excellent CTR (0.960) and CI (0.885) in French individuals with no history of ankle trauma. Rosen et al. [111] showed that the digital version of the CAIT has excellent reliability (0.93). In a study of Chinese patients with two or more sprains, Wang et al. [125] found good CTR (0.930) and satisfactory CI (0.845–0.878). The Thai version (CAIT-THA) showed good CI (0.837) and excellent CTR (2.1) of 0.945 (95% CI = 0.93–0.96) [126]. Khan et al. [127] reported acceptable CI (0.75) and excellent CTR (>0.80) in Pakistani patients with isolated instability. Kunugi et al. [128] observed high CI (0.833) and reliability (ICC = 0.826, 95%; confidence interval: 0.732–0.888) in the Japanese population.

**Table 9 healthcare-12-01733-t009:** Sample composition and subscale scores for a variety of articles using CAIT.

Reference	Sample	Right Ankle	Left Ankle	Total Score
Mirshahi et al. [110]	116 Athletes with and without ankle instability	25.14 (4.98)	25.76 (4.94)	-
Rosen et al. [111]	68 Participants ankle sprain	-	-	21.74 (6.44)
Noronha et al. [112]	131 Participants with ankle sprain	24.7 (5.9)	25.0 (5.9)	-
Kadli et al. [115]	267 Children with and without CAI	24.19 (5.04)	23.51 (4.69)	-
Rodríguez-Fernández et al. [116]	171 Sportsmen with many ankle injuries	CAIT-Sv Right ankle: 25.3 ± 5.3;	CAIT-Sv Left ankle: 26.5 ± 4.3;	-
Ko et al. [117]	168 Olympic sports athlete participants	-	-	24.1 ± 6.8
Hadadi et al. [119]	135 Participants with and without a history of sprains	CAIT Right ankle: 21.71 ± 6.8;	CAIT Left ankle: 24.5 ± 5.5;	-
Vuurberg et al. [120]	130 Participants with symptoms on foot and ankle	-	-	12.35 ± 7.6
Korakakis et al. [121]	107 Athletes with and without tibiotarsal problems and symptoms	-	-	CAI 14.5 (5.7) Lateral ankle sprain 12.4 (7.8) Healthy 29.2 (1.8) Other injury: 27.7 (3)
Lin et al. [122]	135 University athletes with and without CAI	-	-	CAI 16.4 (4.1) Control group 25.6 (4.4)
Tsekoura et al. [123]	123 Participants with and without history of ankle sprain	-	-	Without a history of ankle sprain: 26.9 (3.16) The instability group: 20.6 (4.62)
Kunugi et al. [128]	111 Soccer players with and without CAI	-	-	22.56 ± 4.89

Note. CAIT—Cumberland Ankle Instability Tool; CAI—Chronic Ankle Instability.

#### 3.6.3. Summary

CAIT scores range from 0 to 30, with lower scores indicating more severe instability. Studies show that the questionnaire has favorable psychometric properties, such as good CI and CTR, in different populations and cultures. Most studies reported high CI, with α values above the acceptable cut-off point of 0.70. In addition, CTR was generally considered excellent, with ICC ranging from 0.82 to 0.98. These findings support the use of the CAIT as a reliable tool for assessing FAI in different clinical and sporting contexts. However, Hiller et al. [33] reveal that the fit statistics for 3 questions (general stability, turning, pain) are outside the desired limits. It is important to note that some cultural variations can influence the psychometric properties of the CAIT, which highlights the importance of local validations to ensure its proper applicability and interpretation in different populations.

## 4. Discussion

### 4.1. Main Findings

The present review aimed to analyze the six most cited FAI assessment measures. We considered patient-reported outcome measures related to ankle and foot based in the Model of Chronic Ankle Instability that shows the relationship between the foot and ankle based on Primary Tissue Injury, Pathomechanical Impairments (Pathologic Laxity, Arthrokinematic Restrictions, Osteokinematic Restrictions, Tissue Adaptations), Sensory-Perceptual Impairments (Diminished Somatosensation, Pain, Self-Reported Function), and Motor-Behavioral Impairments (Neuromuscular Inhibition, Altered Reflexes, Muscle Weakness, Balance Deficits, Altered Movement Patterns, Reduced Physical Activity) [1]. Our results showed that the most cited FAI questionnaires are LEFS, FAAM, FFI, FAOS, OMAS, and CAIT. Each questionnaire tries to understand different characteristics, addressing FI in a broad sense. However, there is an important distinction in the construction of each questionnaire. While the LEFS [29] is a specific measure applicable to a wide variety of lower limb orthopedic conditions, the FAAM [23], FFI [30], FAOS [31], OMAS [32], and CAIT [33] are questionnaires developed to assess the functional capacity of the ankle and foot in individuals with injuries or related conditions to this region. It is important to note that this differentiation can provide benefits to a given questionnaire over another. This is due to the possibility that there are essential aspects that the researcher must take into consideration before evaluating the relative advantages of a specific questionnaire, as each questionnaire has its own advantages and is more suitable for specific contexts.

Bibliometric data can also be useful in indicating the impact of using a specific questionnaire [129]. Unlike some critical reviews, such as that by Donahue et al. [27], who highlight CAIT [33] and AII to determine FAI status, or the study by Simon et al. [22], which indicates that it is the Identification of Functional Ankle Instability (IdFAI), our analysis points to the FAOS [31] as the most suitable questionnaire for this field. As previously noted, although the bibliometric analysis revealed that LEFS [29] is the most cited questionnaire in the literature, the FAOS [31] has the greatest overall impact, considering the evaluation of the weighted average impact factor. Thus, this result suggests that FAOS is the questionnaire with the greatest impact in the context of FAI in sport. The FFI [30], in turn, ranks sixth.

There are developmental differences that form two distinct groups among the six questionnaires, based on the sample used in the original publications. Firstly, the LEFS [29], FFI [30], and OMAS [32] are not sport-specific questionnaires, as they were initially used on a sample with different orthopedic conditions. Furthermore, several reduced and adapted versions were developed, which indicates its flexibility and applicability in different contexts. Secondly, the FAAM, FAOS [31], and CAIT [33] were originally non-sporting questionnaires, but comprised questions and sub-scales aimed at sport. Their ability to assess both ADL and sports activities provides a comprehensive view of ankle functionality.

Five of the publications (LEFS, FAAM, FFI, FAOS, OMAS) included samples with fractures, acute or chronic ligament injuries, dysfunctions, and specific foot and ankle pathologies. In contrast, the CAIT [33] was the only one focused on as a LFS assessment tool. The scoring procedures for each questionnaire are straightforward, although the FAAM [23], FFI [30], FAOS [31], and OMAS [32] offer some flexibility, allowing the combination of sub-scale values to provide an overall composite score, the LEFS [29] and CAIT [33] stand out as more simplified instruments, generating a single overall score. This more straightforward approach to scoring is frequently reported in the literature, which contributes to its ease of use and interpretation. As a final comparison, the CAIT [33] is the only one that has a child version available, which is ideal for examining younger samples.

The ease of administration and scoring makes the LEFS [29] an appropriate choice for documenting lower limb function. However, the limitations of its validation study include the predominantly outpatient sample, and it may not be sensitive enough to identify specific problems related to FAI, as it focuses on global lower-limb function, which may limit its usefulness in situations where the assessment needs to be more specific. The factor analysis of their original study indicates that most of the items were related to a single construct, suggesting that the FFI [30] reliably measures foot function. However, the activity limitation subscale showed a slightly lower CI, indicating that the contribution of this measurement subscale is not completely clear. It is important to note that the FFI [30] focuses mainly on foot function and may not be as sensitive to LFS-related problems, being less comprehensive than some other available measures.

The FAOS [31] was originally touted as a useful tool for assessing outcomes related to ankle reconstruction, highlighting its relevance in clinical practice and research. Its ability to measure patient-relevant outcomes is emphasized as an important benefit, contributing to informed treatment choices and identification of costs without benefits. Some limitations are acknowledged, including the fact that it was conducted on a specific group of patients, which limits the generalizability of the results to other populations with ankle problems. The responsiveness of the FAOS [31], i.e., its ability to detect clinical changes over time, was not assessed in its original study, and more research is needed on this aspect. In this way, the FAOS [31] is considered a reliable and valid tool for assessing clinically relevant results in patients undergoing ankle reconstruction; however, some limitations are pointed out in different clinical contexts.

The OMAS [32] scoring system was effective in assessing outcomes after surgical treatment of ankle fractures. It showed strong correlations with several objective parameters, including the patient’s subjective assessment using a linear analog scale, the range of motion in loaded dorsal extension, and the presence of osteoarthritis and ankle dislocations. The OMAS [32] also proved sensitive in detecting differences between groups of patients with different degrees of injury severity. Despite the advantages of the OMAS [32] system, it is important to note that the original article does not specifically mention limitations. However, the study did not evaluate its responsiveness, i.e., how OMAS [32] responds to clinical changes over time. Despite these considerations, the OMAS [32] system is recognized as a valuable tool for assessing clinical outcomes in patients with ankle fractures. Its ability to provide a detailed assessment is emphasized, especially in comparison with broader assessment approaches. Furthermore, it can be applied not only to fractures but also to other ankle-related conditions.

The CAIT [33] presents itself as a reliable and valid measure of FAI, showing a strong correlation with the global perception of ankle instability. However, the Rasch analysis of the CAIT revealed that three questions (general stability, rotation, and pain) were outside the desired limits, suggesting that the CAIT [33] may not be unidimensional for FAI. Despite this, the question about tibiotarsal pain, although considered to be outside the construct of FI, remained in the questionnaire. A notable limitation during the validation of the CAIT [33] was the absence of an established standard criterion for measuring FAI, which made the validation process more challenging. The CAIT [33] demonstrated an ability to identify the severity of FAI by clearly distinguishing between two groups of individuals: those with and without FAI. The CAIT [33] is considered a valuable tool in both clinical and research settings. In the clinic, it can be used to assess the severity of FAI, monitor treatment outcomes, and track patient progress. In research, the possibility exists that it could predict future sprains in individuals with FAI, allowing for a preventative approach. In summary, the CAIT [33] is a simple, reliable, and valid tool for measuring FAI; although, it does present some issues to be considered, such as the possible compression of questions and the need for further investigations into its predictive validity.

### 4.2. Limitations

To our knowledge, this is the first study to identify and analyze the most cited instruments to evaluate FAI in the context of sport. This review could, therefore, aid in developing appropriate interventions that assess FAI and monitor clinical progress over time, particularly in athletes and sportspeople.

As mentioned earlier, there are distinctions between the questionnaires discussed that make them more applicable to certain clinical practices and in research. They have all proved useful in assessing results after different types of ankle interventions. However, some limitations have been identified. The lack of evaluation of responsiveness was a common limitation, meaning that it was not examined how sensitive these measures are to detecting clinical changes over time. This is essential, especially in clinical contexts, where the ability to measure evolution or regression is key. Another common limitation was the absence of a standard criterion for FAI. Lastly, we only included studies written in English, which may have caused us to miss other relevant publications in other languages.

### 4.3. Practical Implications and Suggestions for Future Studies

All the measures have important practical implications. They can be used to assess and monitor treatment progression, as well as to identify more homogeneous groups of patients in research studies. It is essential that health professionals and researchers consider the characteristics and psychometric properties of each questionnaire when selecting the best assessment tool for their clinical and research needs. Therefore, these measures represent valuable tools for healthcare professionals and researchers dealing with ankle-related issues.

Future studies should adopt rigorous and robust methodologies during the validation phase of the instruments used to measure FAI, considering the Assessment of the Reliability and Validity of the Questionnaire, Exploratory Factor Analysis, Confirmatory Factor Analysis, and Multigroup Analysis. These efforts are key for improving the psychometric qualities and standardizing the measure in terms of its application, correction, and interpretation, particularly in the translated versions of the instruments.

## 5. Conclusions

The questionnaires evaluated in this review have different characteristics and are applicable to different populations with FAI in sport. However, the FAOS stands out as the questionnaire with the highest weighted average impact factor, indicating its relevance and recognition in the scientific literature. The use of these measures can make a significant contribution to the accurate and objective assessment of FAI in athletes and sportspeople, helping to propose appropriate interventions and monitor clinical progress over time.

## Figures and Tables

**Figure 1 healthcare-12-01733-f001:**
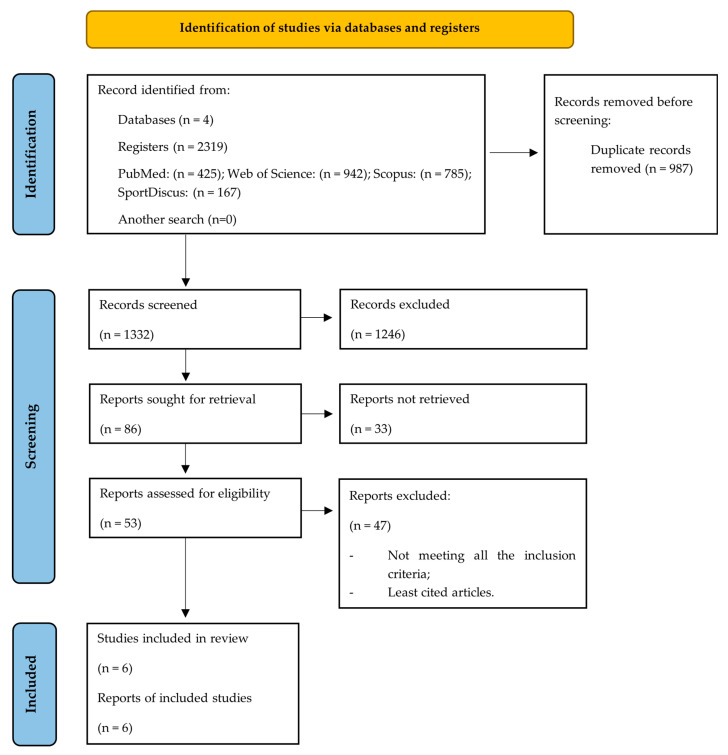
Diagram illustrating each phase of the search and selection process.

**Figure 2 healthcare-12-01733-f002:**
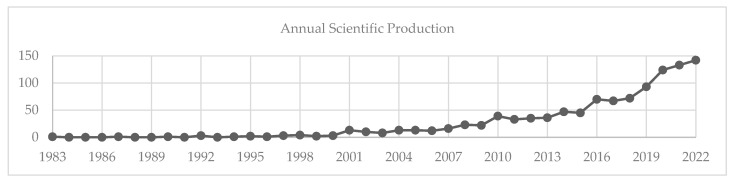
Annual scientific production during 1983–2022.

**Table 1 healthcare-12-01733-t001:** Characteristics of the six most cited evaluation measures.

Measure	Year	Original Authors	Country First Author	Journal	Total Number of Journal Citations *
FAAM	2005	Martin et al. [23]	USA	Foot & Ankle International	10,773
LEFS	1999	Binkley et al. [29]	USA	Physical Therapy	14,922
FFI	1991	Budiman-Mak et al. [30]	USA	Journal of Clinical Epidemiology	41,985
FAOS	2001	Roos et al. [31]	Sweden	Foot & Ankle International	10,773
OMAS	1984	Olerud et al. [32]	Sweden	Archives of Orthopaedic and Trauma Surgery	9889
CAIT	2006	Hiller et al. [33]	Australia	Archives of Physical Medicine and Rehabilitation	25,806

Note. FAAM—Foot and Ankle Ability Measure; LEFS—Lower Extremity Functional Scale; FFI—Foot Function Index; FAOS—Foot and Ankle Outcome Score; OMAS—Olerud and Molander Ankle Score; CAIT—Cumberland Ankle Instability Tool. * These data were collected in July 2024.

**Table 2 healthcare-12-01733-t002:** Overview of the six most cited evaluation measures.

Construct	Measure	Test–Retest Reliability	Items	Subscales	Answers/Items	Classification
Function	FAAM	0.89 and 0.87 for the ADL and Sports subscales	29	2	0–4	0–100
Function	LEFS	0.89	20	-	0–4	0–80
Pain/Restriction/Incapacity	FFI	0.87 to 0.69	23	3	0–9	0–100
Pain/Function	FAOS	0.78, 0.86, 0.70, 0.85, 0.92 for the five subscales	42	5	0–4	0–100
Fracture/Function	OMAS	0.92 to 0.96	9	3	0–3	0–100
Function	CAIT	0.96 to 0.97	9	-	0–5	0–30

Note. FAAM—Foot and Ankle Ability Measure; LEFS—Lower Extremity Functional Scale; ADL—Activities Daily Living; FFI—Foot Function Index; FAOS—Foot and Ankle Outcome Score; OMAS—Olerud and Molander Ankle Score; CAIT—Cumberland Ankle Instability Tool; Items—number of items in the measure; Answers/Items—Likert scale of the measures.

**Table 3 healthcare-12-01733-t003:** Bibliometric data for six highly cited assessment measures in sport.

Measure	Citation	Citations/Year	IF–Journal Original *	Quartile *	Sources with IF		
					Journals	Articles	IF-Weighted Average
LEFS	989	43	3.679	1	310	858	3.59
FAAM	575	33.8	3.569	2	137	508	3.13
FFI	554	17.3	7.407	1	161	483	3.12
FAOS	445	21.2	3.569	2	111	391	4.70
OMAS	391	10.3	2.928	2	86	325	4.42
CAIT	335	19.7	4.06	1	109	322	3.14

Note. Data obtained from “Cited Reference Search” of the Web of Science (Core Collection); IF—impact factor. LEFS—Lower Extremity Functional Scale; FAAM—Foot and Ankle Ability Measure; FFI—Foot Function Index; FAOS—Foot and Ankle Outcome Score; OMAS—Olerud and Molander Ankle Score; CAIT—Cumberland Ankle Instability Tool. * IF—Journal Original and Quartile are referred to 2021.

**Table 5 healthcare-12-01733-t005:** Sample composition and subscale scores for a variety of articles using the FAAM.

Reference	Sample	Subscale ADL	Subscale Sport
Nauck et al. [57]	109 Participants	Conservatively treated CAI patients 91.6 (7.9) Pre-operatory patients 69.3 (20.8) Sport students 99.0 (2.1) Volleyball players 99.0 (1.7)	Conservatively treated CAI 75.4 (14.3) Pre-operatory patients 34.1 (29.0) Sport students 93.5 (15.2) Volleyball players 96.2 (5.5)
Mazaheri et al. [58]	93 Participants of Orthopedic clinics and physiotherapy Iran	69.19 (21.97)	41.67 (25.13)
Bordoz et al. [59]	105 Patients ambulatory	74 (22.1)	44 (31.1)
Uematsu et al. [60]	83 University athletics participants	74.2 (29.4)	52.1 (35.7)
Çelik et al. [61]	98 Participants Department of Orthopedics and Traumatology at the University of Medipol	First evaluation 57.5 (95% CI = 51.6–65.2) Second evaluation 55.1 (95% CI = 48.6–62.6)	First evaluation 19.4 (95% CI = 16.2–22.4) Second evaluation 18.4 (95% CI = 15.6–22.9)
Moreira et al. [62]	90 Participants Private Clinic	70.72 (19.36)	38.13 (27.59)
Arnold et al. [63]	68 Patients university and city population	Functional ankle instability group 93.71 (6.15) Uninjured group 99.51 (1.35)	Functional ankle instability group 84.47 (8.40) Uninjured group 99.78 (0.72)

Note. FAAM—Foot and Ankle Ability Measure; ADL—activities daily living.

**Table 6 healthcare-12-01733-t006:** Sample composition and subscale scores for a variety of articles using the FFI.

Reference	Sample	Pain	Disability	Activity Limitation	Total Score
Budiman-Mak et al. [30]	87 Participants	29.71 (28.13)	41.36 (30.74)	14.94 (19.13)	28.09 (23.26)
Wu et al. [72]	88 Participants	50.9 (24.6)	40.6 (26.5)	21.9 (19.8)	38.5 (18.9)
Martinelli et al. [73]	89 Participants	First evaluation 55.9 (24.8) Second evaluation 56.9 (24.2)	First evaluation 48.8 (28.8) Second evaluation 50.8 (27.7)	-	-
Jorgensen et al. [74]	35 Participants	63 (9)	70 (8)	2 (median) and 0–3 (interquartile range)	135 (12)
Martinez et al. [75]	50 Participants	30.39 (24.94)	12.34 (12.91)	18.89 (14.22)	20.54 (15.17)
Külünkoğlu et al. [76]	159 Participants	20.65 (10.43)	19.79 (14.04)	4.67 (5.82)	45.11 (27.09)

Note. FFI—Foot Function Index.

## Data Availability

All data supporting were included on this paper.

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
