# Peer review of "Assessing Functional Ankle Instability in Sport: A Critical Review and Bibliometric Analysis"

_healthcare, 2024, doi:10.3390/healthcare12171733_

Round 1

Reviewer 1 Report (New Reviewer)

Comments and Suggestions for Authors

Thank you for inviting me to review this analysis on the most cited scales related to assessing functional ankle instability in sports. In my opinion, authors present a well-prepared manuscript that offers a clear orientation on orienting researchers, however, I have some comment that help improve the work:

1- The abstract would benefit from mentioning that 4 databases were searched.

2- The objectives in the last paragraph need to be more clear and straight forward. I suggest elaboration

3- The first paragraph of the materials and methods need to have a subtitle. I suggest "search strategy".

4- It is not clear how authors found the most cited articles from all the results reached after the search.

5- The authors used some abbreviations, then they used full words. This must be consistent all over the manuscript, including the conclusion.

6- The conclusion well fits the manuscript.

All in all, this manuscript offers an addition to the literature, and is interesting for readers.

I suggest acceptance after some minor changes.

Good luck

Author Response

Dear Editor

RE: Measures for Assessing Functional Ankle Instability in Sport: A Critical Review and Bibliometric Analysis.

My colleagues and I would like to thank you for the opportunity to resubmit our manuscript to Healthcare. We found reviewers’ comments to be very helpful, and we have done our best to incorporate all their suggestions. We believe that this has made a significant contribution to the overall quality of the manuscript.

The reviewers’ comments and our actions are attached at the bottom of this letter. We have also included an updated version of our manuscript with all the changes highlighted in track-changes MS-word. We also rewrote some sentences in the manuscript (highlighted in yellow) according to the Editor’s comment.

If you require any additional information, please do not hesitate to get in touch with us.

Thank you for considering our manuscript.

Yours sincerely,

Pedro Mendes, PhD

RESPONSE TO REVIEWER 1

Thank you for inviting me to review this analysis on the most cited scales related to assessing functional ankle instability in sports. In my opinion, authors present a well-prepared manuscript that offers a clear orientation on orienting researchers, however, I have some comment that help improve the work:

Authors' response: Thank you for the valuable suggestions for improving our paper. We have studied comments carefully and have made corrections which we hope meet with approval.

1- The abstract would benefit from mentioning that 4 databases were searched.

Authors' response: We are thankful for the reviewer’s comment.

Action: We have added the four searched databases in the abstract:

Lines 19-20: “Four electronic databases (Web of Science, Scopus, Pubmed and SportDiscus) were searched (up to November 2022)”.

2- The objectives in the last paragraph need to be more clear and straight forward. I suggest elaboration

Authors' response: We are thankful for the reviewer’s comment. We have restructured the last paragraph of the introduction.

Action: For clarity, we have removed some redundant information in Lines 93-95 and divided the mentioned paragraph into two, to give more focus to the objective of the study.

Line 98-100: “Therefore, the purpose of this study is to identify, overview and evaluate the six self-report measures most frequently used to assess FAI, with a specific focus on patient-self-reported outcome measures related to ankle and foot.”

3- The first paragraph of the materials and methods need to have a subtitle. I suggest "search strategy".

Authors' response: We agree with the reviewer's comment.

Action: We added the suggested subtitle in Line 102: “2.1. Search Strategy”.

In the same sense, we have updated the numeration of the remaining subtitles.

4- It is not clear how authors found the most cited articles from all the results reached after the search.

Authors' response: Thank you for your important comment. The six most cited questionnaires were found using the Cited Reference Search in Web of Science.

Action: To the following information: “The six most cited LFS evaluation questionnaires were selected for analysis”, presented in Line 161, we added the following information: “… using the Cited Reference Search in Web of Science…”.

5- The authors used some abbreviations, then they used full words. This must be consistent all over the manuscript, including the conclusion.

Authors' response: We are thankful for the reviewer’s important comment.

Action: We have revised the manuscript to correct these mistakes.

6- The conclusion well fits the manuscript.

All in all, this manuscript offers an addition to the literature, and is interesting for readers.

I suggest acceptance after some minor changes.

Good luck.

Authors' response: Thank you for your comment. We did our best to incorporate all your suggestions and to respond to comments. We believe that the changes made substantially improved the manuscript. We included an updated version of our Word manuscript with all the changes highlighted. Thank you very much for your attention and suggestions on the manuscript.

Reviewer 2 Report (New Reviewer)

Comments and Suggestions for Authors

Thank you for the opportunity to examine this review and bibliometric analysis of the six most cited FAI assessment measures.

I consider it necessary to improve the presentation of the study.

METHODS

-How are the 6 most cited questionnaires for assessing FAI identified?

-The search strategy could be improved by truncating the terms correctly. Terms composed of more than two words should always be written in quotation marks. I believe that the appropriate approach to this strategy could have retrieved different documents, so it is advisable to include it in the limitations of the study.

-In selection criteria: vi) Studies with Patient Reported Outcome Measures related to ankle and foot measures” add “self-reported”.

-2.2. Data extraction

As stated, the data extraction was done by one person. What was done independently was the selection of articles. Correct this aspect.

-Indicate whether any software or other tool has been used to eliminate duplicates, to perform the peer review, and for data extraction.

-“extracted the following information from the primary studies: name of the measure, first author, year of publication, information about the scale, number of citations, number of citations per year, impact factor of the original journal, number of publications, number of different journals, information about the scale and sample”. More data is extracted and reflected in the tables. Complete.

-The flow chart contains several errors. There are more than 3 databases. The exclusion of documents is not adequately justified. Finally, the 6 most cited questionnaires are included, but that is not the number of studies in the review. Review the calculations and express it correctly.

-“Figure 2. Annual Scientific Production During 1983-2022” should appear as a caption.

The selection criteria do not determine any publication date. It is necessary to justify the period 1983-2022.

-LINES 183-186: what is expressed in this paragraph is not sufficiently clear. It should be clarified whether at any time during the study, any type of selection by journal or by discipline has been made.

-“Table 1. Characteristics of the six most cited evaluation measures”. It is not clear when (at what time) the Total number of journal citations is computed.

-“Table 2. Overview of the six most cited evaluation measures”. Add ADL in the legend.

-“Table 3. Bibliometric data for six highly cited assessment measures in sport”. Review data and always put a period, or always a comma. The data on the quartile and impact factor, to which year do they refer?

-3.1.1. Development

The LEFS (Appendix IV). Where are the appendices? Why do they start with Appendix IV?

- Always reference this test in the same way “the LEFS 15 item”, “the 15-item LEFS”.

- Make sure that all abbreviations are spelled out in the text the first time, for example: CI, CTR, MMD, IFT….

-“Table 5. Sample composition and subscale scores for a variety of articles using the FAAM”. Check “conservative patients”, “Study sports”, “No injury”.

-“Table 7. Sample composition and subscale scores for a variety of articles using the FAOS.” Check the header and legend. (ADL, QoL – Quality of Life). Check the study by Sierevelt et al. [81].

-“Tables 4 and 8. Sample composition and subscale scores for a variety of articles using the OMAS.”. In these cases, it may not be necessary to talk about subscale scores.

- It is suggested that the studies be ordered in the Tables using a certain criterion (alphabetical order, publication date, citation order...). I believe that all the Tables should be reviewed and given a more uniform format in terms of headings, upper and lower case letters, semicolons, order of the studies, legends of the Tables, etc.

DISCUSSION

-Highlight the strong points of this study. In the limitations, discuss any limitations of the review processes used and any limitations of the evidence included in the review.

Add possible future lines of research.

REFERENCES

Abbreviate the journal title in all cases.

Author Response

Dear Editor

RE: Measures for Assessing Functional Ankle Instability in Sport: A Critical Review and Bibliometric Analysis.

My colleagues and I would like to thank you for the opportunity to resubmit our manuscript to Healthcare. We found reviewers’ comments to be very helpful, and we have done our best to incorporate all their suggestions. We believe that this has made a significant contribution to the overall quality of the manuscript.

The reviewers’ comments and our actions are attached at the bottom of this letter. We have also included an updated version of our manuscript with all the changes highlighted in track-changes MS-word. We also rewrote some sentences in the manuscript (highlighted in yellow) according to the Editor’s comment.

If you require any additional information, please do not hesitate to get in touch with us.

Thank you for considering our manuscript.

Yours sincerely,

Pedro Mendes, PhD

RESPONSE TO REVIEWER 2

Thank you for the opportunity to examine this review and bibliometric analysis of the six most cited FAI assessment measures.

I consider it necessary to improve the presentation of the study.

Authors' response: Thank you for the valuable suggestions for improving our paper. We have studied comments carefully and have made corrections which we hope meet with approval.

METHODS

-How are the 6 most cited questionnaires for assessing FAI identified?

Authors' response: We are thankful for the reviewer’s comment. The six most cited questionnaires are identified using the Cited Reference Search in Web of Science platform.

Action: We added the following information in lines 161-162: “… questionnaires were selected for analysis using the Cited Reference Search in Web of Science and…”.

-The search strategy could be improved by truncating the terms correctly. Terms composed of more than two words should always be written in quotation marks. I believe that the appropriate approach to this strategy could have retrieved different documents, so it is advisable to include it in the limitations of the study.

Authors' response: We are thankful for the reviewer’s comment. We reviewed our search strategy and verified that we used quotation marks for terms made up of two or more words. We therefore assume that it was a typo here in the text of the manuscript. Thank you very much for picking up on this.

Action: We have put in the correct search strategy that we used in the four electronic databases (Lines 114-118): “By including the boolean indicators "AND" or "OR", the search strategy resulted in the following: ((ankle OR “ankle joint” OR “ankle injury” OR “functional instability”) AND (sport* or exercise* OR training) AND (survey OR questionnaire* OR instrument* OR measur* OR scale OR "subjective evaluation" OR assess* OR tool OR "self-report*") AND ("validation studies" OR psychometr* OR valid* OR "internal consistency")).”

-In selection criteria: vi) Studies with Patient Reported Outcome Measures related to ankle and foot measures” add “self-reported”.

Authors' response: Thank you for your comment. As suggested, we added the term “self-reported” in the sentence.

Action: We have added the term “self-reported” in the line 127 (eligibility criteria).

-2.2. Data extraction

As stated, the data extraction was done by one person. What was done independently was the selection of articles. Correct this aspect.

Authors' response: We are thankful for the reviewer’s important comment. The selection of the studies and data extraction were carried out by two independent reviewers. We have corrected this point in the text.

Action: We have corrected the sentences in Lines 132-134: “Two authors carried out the search, selection of the articles, and data extraction independently and the results were later compared and discussed. The authors extracted the following…”.

-Indicate whether any software or other tool has been used to eliminate duplicates, to perform the peer review, and for data extraction.

Authors' response: We are thankful for the reviewer’s important comment. The studies were imported into the software EndNote v21.1 and duplicates were removed.

Action: We have added the following sentence in lines 143-144: “EndNote v21.1 software was used for peer review, elimination of duplicate articles and data extraction”.

-“extracted the following information from the primary studies: name of the measure, first author, year of publication, information about the scale, number of citations, number of citations per year, impact factor of the original journal, number of publications, number of different journals, information about the scale and sample”. More data is extracted and reflected in the tables. Complete.

Authors' response: Thank you for your important comment. We have updated the extracted information from the primary studies.

Action: We have updated the information in lines 135-140: “name of the measure, first author, year of publication, country first author, journal where it was published, total number of citations of the journal, information about the scale including test-retest reliability, items, subscales, answers/items, classification, number of citations, citations/years, impact factor of original journal, quartile, number of journals that cited it, number of manuscripts that cited the measures and impact factor weighted average”.

-The flow chart contains several errors. There are more than 3 databases. The exclusion of documents is not adequately justified. Finally, the 6 most cited questionnaires are included, but that is not the number of studies in the review. Review the calculations and express it correctly.

Authors' response: We are thankful for the reviewer’s comment. We are very sorry for the typos in Figure 1. These numbers have been revised and corrected.

In relation to the justification of excluded documents, previous works (Page et al., The BMJ, 2021) only suggests that authors report the number of records excluded after screening titles or titles and abstracts. For this reason, we have not listed the specific reasons for excluding the reports based on the title/abstract (as we did with the full text). However, in general, the reports excluded at these stages did not present Patient-Self-Reported Outcome Measures related to ankle and foot measures or did not present validation data.

As mentioned in the Methods section, the flowchart only shows the result of the search carried out to find the six most cited measures to evaluate FAI. These results are shown in Tables 1-3. The information in Tables 4-9 were articles that cited these original measures found in the flowchart process.

Action: A new flow chart has been inserted. Please, consider it.

“Figure 2. Annual Scientific Production During 1983-2022” should appear as a caption.

Authors' response: We are thankful for the reviewer’s comment.

Action: We have correctly repositioned the figure caption. Please, see line 159.

The selection criteria do not determine any publication date. It is necessary to justify the period 1983-2022.

Authors' response: Thank you for your important comment. Methodologically we have chosen not to include any period of retreat, to cover as many potential measures as possible.

Action: We have added the following information in Line 108: “…considering the maximum period of retreat, in order to…”.

-LINES 183-186: what is expressed in this paragraph is not sufficiently clear. It should be clarified whether at any time during the study, any type of selection by journal or by discipline has been made.

Authors' response: Dear reviewer, no selection was made by journal or by discipline.

Action: We have added the following information in Lines 198-199: “Similar to Clancy et al. [28] study, no selection was made by journal or discipline.”

-“Table 1. Characteristics of the six most cited evaluation measures”. It is not clear when (at what time) the Total number of journal citations is computed.

Authors' response: Thank you for your important comment. The total number of journal citations presented in Table 1 was computed this year, according to the suggestions of another reviewer.

Action: We have added the following information to the Notes of Table 1: “* This data was collected in July 2024”.

-“Table 2. Overview of the six most cited evaluation measures”. Add ADL in the legend.

Authors' response: Thank you for your important comment.

Action: We added the meaning of the acronym ADL (i.e., Activities daily Living) in the legend of Table 2.

-“Table 3. Bibliometric data for six highly cited assessment measures in sport”. Review data and always put a period, or always a comma. The data on the quartile and impact factor, to which year do they refer?

Authors' response: Thank you for your important comment. We have replaced the commas with periods. The quartile and impact factor data refer to the year 2021, as mentioned in the Methods section.  

Action:

All commas presented in Table 3 were replaced with periods.

We have added the following sentence in the Notes of Table 3: “IF-Journal Original and Quartile are referred to 2021”.

-3.1.1. Development

The LEFS (Appendix IV). Where are the appendices? Why do they start with Appendix IV?

Authors' response: We are thankful for the reviewer’s comment. There are no appendices in our article, therefore this indication represents an error. In our initial version of the manuscript, we had the six most cited questionnaires displayed in the respective appendices, however, we opted to exclude it.

Action: We have removed the indication of the Appendices from the manuscript.

- Always reference this test in the same way “the LEFS 15 item”, “the 15-item LEFS”.

Authors' response: Thank you for your important comment.

Action: As suggested, we have referenced the 15-item LEFS in the same way throughout the manuscript.

- Make sure that all abbreviations are spelled out in the text the first time, for example: CI, CTR, MMD, IFT….

Authors' response: Thank you for your important comment.

Action: We have spelled out all the abbreviation in the text for the first time.

Line 242: CI, confidence interval; CTR, test-retest reliability (CTR)

Line 268-269: MMD, minimum detectable change

Line 519: IFT is incorrect and was replaced by FAI.

-“Table 5. Sample composition and subscale scores for a variety of articles using the FAAM”. Check “conservative patients”, “Study sports”, “No injury”.

Authors' response: Thank you for your important comment. We agree with the reviewer, and we have reworded the terms presented for greater clarity.

Action:

The term “conservative patients” was replaced by “conservatively treated CAI patients”. 

The term “Study sports” was replaced by “Sport students”

The term “No injury” was replaced by "Uninjured group”.

-“Table 7. Sample composition and subscale scores for a variety of articles using the FAOS.” Check the header and legend. (ADL, QoL – Quality of Life). Check the study by Sierevelt et al. [81].

Authors' response: Thank you for your careful review. We have revised the table and corrected the terms indicated.

Action: We have corrected the header and legend of Table 7, as well as the sample of the study by Sierevelt et al. [81].

-“Tables 4 and 8. Sample composition and subscale scores for a variety of articles using the OMAS.”. In these cases, it may not be necessary to talk about subscale scores.

Authors' response: We are thankful for your suggestion. However, the authors decided to maintain this information to give more information/knowledge to the readers about the subscales of each instrument.

- It is suggested that the studies be ordered in the Tables using a certain criterion (alphabetical order, publication date, citation order...). I believe that all the Tables should be reviewed and given a more uniform format in terms of headings, upper and lower case letters, semicolons, order of the studies, legends of the Tables, etc.

Authors' response: Thank you very much. We agree with the reviewer that it is important to organize studies in the Tables using certain criteria. The studies in Table 3 were ordered according to the number of citations (from the highest to the lowest number), as mentioned in lines 203-204. All other tables were organized according to the order of citation in the text. We also review and standardize the tables, considering titles, semicolons, etc.

Action: We review all the tables and organize the studies. Please consider all the changes made to Tables 1 to 9.

DISCUSSION

-Highlight the strong points of this study. In the limitations, discuss any limitations of the review processes used and any limitations of the evidence included in the review.

Authors' response: We are thankful for the reviewer’s comment. As suggested, we highlighted the strengths of this study in the Discussion section. In the review process, as previously mentioned, we revised our search strategy and verified that we used quotation marks for terms made up of two words. We therefore assume that it was a typo in the text of the manuscript, that was already corrected. For this reason, we do not raise any limitations related to the search strategy. However, it is important to note that we only included studies written in English in our review, which could potentially overlook other relevant publications written in other languages. We considered it as a limitation of this study.

Action:

Line 689-691: “To our knowledge, this is the first study to identify and analyze the most cited instruments to evaluate FAI in the context of sport. This review could therefore aid in developing appropriate interventions that assess FAI and monitor clinical progress over time, particularly in athletes and sportspeople.”

Line 700-702: “Lastly, we only included studies written in English, which may have caused us to miss other relevant publications in other languages.

Add possible future lines of research.

Authors' response: We are thankful for the reviewer’s comment. As suggested, we added some suggestions for futures studies.

Action: We added the following suggestions for future studies in lines 713-718: “Future studies should adopt rigorous and robust methodologies during the validation phase of the instruments used to measure FAI, considering the Assessment of the Reliability and Validity of the Questionnaire, Exploratory Factor Analysis, Confirmatory Factor Analysis, and Multigroup Analysis. These efforts are key for improving the psychometric qualities and standardizing the measure in terms of its application, correction, and interpretation, particularly in the translated versions of the instruments.”

REFERENCES

Abbreviate the journal title in all cases.

Authors' response: We are thankful for the reviewer’s comment. We have improved this section based on the reviewers' suggestions.

Action: We abbreviate the journal title in all cases.

Dear reviewer, thank you very much for your careful review and suggestions. We did our best to incorporate all your suggestions and to respond to comments. We believe that the changes made substantially improved the manuscript. We included an updated version of our Word manuscript with all the changes highlighted.

Round 2

Reviewer 2 Report (New Reviewer)

Comments and Suggestions for Authors

Thank you for the authors’ effort to address the previous comments. The revised manuscript has been improved.

This manuscript is a resubmission of an earlier submission. The following is a list of the peer review reports and author responses from that submission.

Round 1

Reviewer 1 Report

Comments and Suggestions for Authors

1. The study is overall good with scientific evidence, however, some defects found below:

-line 121 "(OMAS; [28]) e Cumberlang", shoulder be ", Cumberland"

-table 1, "Itens*" and "Answers / Itens" what do they mean?

-table 2, "FI-Journal Original", "Sources with FI", "FI Weighted

Average",  what do they mean?

-line 328, "3.4. Foot Function Index", do you mean "FAOS"?

2. the content of "Reliability" of each score is redundant with difficulty to read. Please simplify for increasing readability. 

Comments on the Quality of English Language

Moderate editing of English language required

Author Response

Dear Ms. Amy Su

RE: Measures for Assessing Functional Ankle Instability in Sport: A Critical Review and Bibliometric Analysis.

My colleagues and I would like to thank you for the opportunity to resubmit our manuscript to Healthcare. We found reviewers’ comments to be very helpful, and we have done our best to incorporate all their suggestions. We believe that this has made a significant contribution to the overall quality of the manuscript.

The reviewers’ comments and our actions are attached at the bottom of this letter. We have also included an updated version of our manuscript with all the changes highlighted in track-changes MS-word.

If you require any additional information, please do not hesitate to get in touch with us.

Thank you for considering our manuscript.

Yours sincerely,

Pedro Mendes, PhD

RESPONSE TO REVIEWER 1

  1. The study is overall good with scientific evidence, however, some defects found below:

Authors' response: Thank you for the valuable suggestions for improving our paper. We have addressed each of your comments and revised the manuscript, highlighting all the changes with the tool “track-changes MS-word”.

-line 121 "(OMAS; [28]) e Cumberlang", shoulder be ", Cumberland"

Authors' response:  Dear reviewer, thank you for your comment. We have corrected the typo.

Action: We have corrected the typo error “OMAS; [28]) e Cumberlang” to “OMAS; [28] and Cumberland …” (line 128).

-table 1, "Itens*" and "Answers / Itens" what do they mean?

Authors' response: Dear reviewer, “Items” corresponds to the number of items in the measure and “Responses/Items” corresponds to the Likert scale of the measures.

Action: We added this information in the notes on the Table 1.

-table 2, "FI-Journal Original", "Sources with FI", "FI Weighted Average",  what do they mean?

Authors' response: Dear reviewer, thank you for your comment. We have corrected “FI” to “IF” (i.e., impact factor). Thus, IF-Journal Original corresponds to the impact factor of each journal, where the original articles were published. Sources with IF correspond to published literature with an impact factor. The weighted average IF, as the name implies, is the weighted average impact factor. Lines 135-139 explain how to calculate it. More information, including the meaning of the abbreviation IF, can be found in the notes of the Table 2.

-line 328, "3.4. Foot Function Index", do you mean "FAOS"?

Authors' response: We are thankful for the reviewer’s comment. The topic 3.4. is related to “Foot and Ankle Outcome Score” not “Foot Function Index”.

Action: We have changed “3.4. Foot Function Index” to “3.4. Foot and Ankle Outcome Score”.

  1. the content of "Reliability" of each score is redundant with difficulty to read. Please simplify for increasing readability.

Authors' response: Dear reviewer, thank you for your comment. We intend to present all the validations we found in the literature and explore the original study a little. If you wish, we can leave only the information presented in table 4.

Reviewer 2 Report

Comments and Suggestions for Authors

Thank you for the opportunity to review this manuscript, and I hope these suggestions are helpful for its improvement. The manuscript presents a critical review and bibliometric analysis of measures for assessing functional ankle instability (FAI) in sports. While the topic is relevant, the objective is not clearly defined, and several methodological and structural issues reduce the overall quality and impact of the study. The following critical points need to be addressed to significantly improve the manuscript:

Clarity of Objective: The objective of the study is not clearly defined. The authors should more precisely explain what they aim to achieve with this review and bibliometric analysis.

Selection of PROMs (Patient-Reported Outcome Measures): The article mentions PROMs related to ankle instability but then selects general foot and ankle PROMs. This selection needs a stronger justification. Only questionnaires specifically used in studies on ankle instability should be considered.

Bibliometric Data: The inclusion of basic bibliometric data such as Impact Factor and citations seems insufficient to justify the selection of certain PROMs. A deeper and more contextualized analysis of these data is necessary.

Inconsistencies in the Flow Diagram: The flow diagram mentions 6 selected articles, but many more appear in the tables. This inconsistency should be corrected to ensure coherence and transparency in the study selection process.

Relevance and Conciseness of Content:The text is lengthy and contains many irrelevant data, often unrelated to ankle instability. It is crucial to focus the content on questionnaires specifically used in studies of ankle instability and analyze their results in detail.

As an example, refer to the 2022 article available on SAGE Journals, which addresses a similar approach to that proposed by the authors but more clearly and specifically. This article can serve as a guide to improve the structure and focus of the presented work.

In summary, for the manuscript to be considered for publication, it is essential that the authors: Clearly define the study's objective. Focus only on questionnaires used in studies on ankle instability. Adequately justify the selection of PROMs with a deeper bibliometric analysis. Correct inconsistencies in the flow diagram and tables. Remove irrelevant information to make the text more concise and focused.

Author Response

Dear Ms. Amy Su

RE: Measures for Assessing Functional Ankle Instability in Sport: A Critical Review and Bibliometric Analysis.

My colleagues and I would like to thank you for the opportunity to resubmit our manuscript to Healthcare. We found reviewers’ comments to be very helpful, and we have done our best to incorporate all their suggestions. We believe that this has made a significant contribution to the overall quality of the manuscript.

The reviewers’ comments and our actions are attached at the bottom of this letter. We have also included an updated version of our manuscript with all the changes highlighted in track-changes MS-word.

If you require any additional information, please do not hesitate to get in touch with us.

Thank you for considering our manuscript.

Yours sincerely,

Pedro Mendes, PhD

RESPONSE TO REVIEWER 2

Thank you for the opportunity to review this manuscript, and I hope these suggestions are helpful for its improvement. The manuscript presents a critical review and bibliometric analysis of measures for assessing functional ankle instability (FAI) in sports. While the topic is relevant, the objective is not clearly defined, and several methodological and structural issues reduce the overall quality and impact of the study. The following critical points need to be addressed to significantly improve the manuscript:

Authors' response: Thank you for the valuable suggestions for improving our paper. We have addressed each of your comments and revised the manuscript, highlighting all the changes with the tool “track-changes MS-word”.

Clarity of Objective: The objective of the study is not clearly defined. The authors should more precisely explain what they aim to achieve with this review and bibliometric analysis.

Authors' response: We are thankful for the reviewer’s comment. As suggested, we have improved the objective of this study.

Action: We reformulated our objective in line with the reviewer's comments (lines 76-78): “Therefore, the purpose of this study is to identify, overview and evaluate the 6 self-report measures most frequently used to assess FAI in bibliometric terms.”

Selection of PROMs (Patient-Reported Outcome Measures): The article mentions PROMs related to ankle instability but then selects general foot and ankle PROMs. This selection needs a stronger justification. Only questionnaires specifically used in studies on ankle instability should be considered.

Authors' response:  Thank you for your comment.  The question you raise is very pertinent and was an element of discussion by the authors of the article. As mentioned in the article, the main objective of this study was to review and evaluate the six self-report measures most frequently used to assess Functional Ankle Instability, following a comprehensive search of the relevant literature in the field, utilizing the bibliometric methods proposed by Clancy et al. (2017). To that end, we endeavored to scrupulously adhere to the methods described in the article, and as you can see, the term "foot" was not introduced. Therefore, we decided to consider ankle and foot PROMs based on the logical reasoning of several factors, taking into account that ankle instability is defined as an impairment in proprioception, neuromuscular control, postural control, and strength (Delahunt et al., 2010), and is characterized by repetitive episodes or perceptions of the ankle giving way, ongoing symptoms such as pain, weakness, reduced ankle range of motion (ROM), diminished self-reported function, and recurrent ankle sprains that persist for more than one year after the initial injury (Gribble et al., 2013). Based on these assumptions, we aimed to follow the Model of Chronic Ankle Instability (Hertel and Corbett, 2019) with an underpinning of the biopsychosocial model, the concepts of self-organization, and perception-action cycles. This model underscores the importance of the foot in understanding functional instability of the ankle, based on Primary Tissue Injury, Pathomechanical Impairments (Pathologic Laxity, Arthrokinematic Restrictions, Osteokinematic Restrictions, Tissue Adaptations), Sensory-Perceptual Impairments (Diminished Somatosensation, Pain, Self-Reported Function), and Motor-Behavioral Impairments (Neuromuscular Inhibition, Altered Reflexes, Muscle Weakness, Balance Deficits, Altered Movement Patterns, Reduced Physical Activity). These were the main points we considered in deciding which PROMs to include in the questionnaires. We will improve the text and sincerely hope you can understand our decision.

Clancy, R. B., Herring, M. P., & Campbell, M. J. (2017). Motivation Measures in Sport: A Critical Review and Bibliometric Analysis. Frontiers in Psychology, 8. https://doi.org/10.3389/fpsyg.2017.00348

Delahunt, E., Coughlan, G. F., Caulfield, B., Nightingale, E. J., Lin, C., and Hiller, C.E. (2010) Inclusion Criteria When Investigating Insufficiencies in Chronic Ankle Instability. Medicine and Science in Sports and Exercise, 42, 2106–2121. https://doi.org/10.1249/MSS.0b013e3181de7a8a

Gribble P. A, et al. (2016) Consensus statement of the International Ankle Consortium: prevalence, impact and long-term consequences of lateral ankle sprains. British Journal of Sports Medicine, 50(24):1493–1495. https://doi.org/10.1136/bjsports-2016-096188

Hertel, J. and Corbett, R. (2019). An Updated Model of Chronic Ankle Instability. Journal of Athletic Training, 54(6): 572-588. https://doi.org/10.4085/1062-6050-344-18

Action: We have added the following information in the “Methods” and “Discussion” sections:

Lines 102-103: “vi) Studies with Patient-Reported Outcome Measures related to ankle and foot measures.”

Lines 535-542: “It was considered Patient-Reported Outcome Measures related to ankle and foot based in the Model of Chronic Ankle Instability that shows the relation between foot and ankle based on Primary Tissue Injury, Pathomechanical Impairments (Pathologic Laxi-ty, Arthrokinematic Restrictions, Osteokinematic Restrictions, Tissue Adaptations), Sensory-Perceptual Impairments (Diminished Somatosensation, Pain, Self-Reported Function), and Motor-Behavioral Impairments (Neuromuscular Inhibition, Altered Re-flexes, Muscle Weakness, Balance Deficits, Altered Movement Patterns, Reduced Physical Activity) [1].”

Bibliometric Data: The inclusion of basic bibliometric data such as Impact Factor and citations seems insufficient to justify the selection of certain PROMs. A deeper and more contextualized analysis of these data is necessary.

Authors' response: We are thankful for the reviewer’s comment. We believe that a bibliometric analysis is useful for discovering emerging trends in the performance of articles and journals, collaboration patterns, and research constituents, and to explore the intellectual structure of a specific domain in the extant literature. The data that claim centre stage in bibliometric analysis tend to be massive and objective in nature (e.g. number of citations and publications, occurrences of keywords and topics), although their interpretation often depends on objective (e.g. performance analysis) and subjective (e.g. thematic analysis) evaluations established through informed techniques and procedures. In other words, bibliometric analysis is useful for deciphering and mapping the cumulative scientific knowledge and evolutionary nuances of well-established fields, making sense of large volumes of unstructured data in rigorous ways. Therefore, well-done bibliometric studies can build solid foundations for advancing a field in new and meaningful ways - they enable and empower academics to (1) gain a complete overview, (2) identify knowledge gaps, (3) derive new ideas for investigation and (4) position their intended contributions to the field (Donthu et al., 2021; Öztürk et al., 2024). It was only with this aim in mind that we went ahead with this study.

Donthu, N., Kumar, S., Mukherjee, D., Pandey, N., & Lim, W. M. (2021). How to conduct a bibliometric analysis: An overview and guidelines. Journal of Business Research, 133, 285–296. https://doi.org/10.1016/j.jbusres.2021.04.070

Öztürk, O., Kocaman, R., & Kanbach, D. K. (2024). How to design bibliometric research: An overview and a framework proposal. Review of Managerial Science. https://doi.org/10.1007/s11846-024-00738-0

Inconsistencies in the Flow Diagram: The flow diagram mentions 6 selected articles, but many more appear in the tables. This inconsistency should be corrected to ensure coherence and transparency in the study selection process.

Authors' response: Thank you for your comment. As we mentioned in the results section (lines 118-121), the flow diagram only shows the result of the search carried out to find the 6 measures most used to identify FAI. Thus, the information in Tables 3-8 were articles that cited these original measures found in the flow diagram process.

Relevance and Conciseness of Content: The text is lengthy and contains many irrelevant data, often unrelated to ankle instability. It is crucial to focus the content on questionnaires specifically used in studies of ankle instability and analyze their results in detail.

Authors' response: Dear reviewer, the results of our study are shown in Tables 1 and 2. The rest of the content is a presentation of studies, and their contexts, which have used the measures identified. This approach has been used in other bibliometric analysis (i.e., Clancy et al., 2017).

Clancy, R. B., Herring, M. P., & Campbell, M. J. (2017). Motivation Measures in Sport: A Critical Review and Bibliometric Analysis. Frontiers in Psychology, 8. https://doi.org/10.3389/fpsyg.2017.00348

As an example, refer to the 2022 article available on SAGE Journals, which addresses a similar approach to that proposed by the authors but more clearly and specifically. This article can serve as a guide to improve the structure and focus of the presented work.

Authors' response: Dear reviewer, please mention the article so that we can improve ours. We are available and interested in making any changes or reformulations that will improve the quality of the article.

In summary, for the manuscript to be considered for publication, it is essential that the authors: Clearly define the study's objective. Focus only on questionnaires used in studies on ankle instability. Adequately justify the selection of PROMs with a deeper bibliometric analysis. Correct inconsistencies in the flow diagram and tables. Remove irrelevant information to make the text more concise and focused.

Authors' response: Dear reviewer, thank you for all your comments and suggestions for improvement. The study objective has been reformulated. As for the rest, we think some confusion has been created because, as we mentioned earlier, the results of our study are only mentioned in table 1 and 2. We hope we have clarified all these issues.

Reviewer 3 Report

Comments and Suggestions for Authors

Functional ankle instability (FAI) is widely researched in sports. This review examines the six most frequently cited questionnaires for assessing FAI: the Lower Extremity Functional Scale (LEFS), Foot and Ankle Ability Measure (FAAM), Foot Function Index (FFI), Foot and Ankle Outcome Score (FAOS), Olerud and Molander Ankle Score (OMAS), and Cumberland Ankle Instability Tool (CAIT). Each tool was evaluated for development, reliability, and overall summary. Bibliometric analysis showed that FAOS ranks highest and FFI lowest in the impact factor of their original publications. Despite variations in psychometric properties, all six questionnaires are robust and well-supported in the literature. Researchers should select tools based on psychometric strengths, limitations, and specific research needs, considering the impact factor as a relevant metric.

The study is of interest and provides beneficial information for those searching for questionnaires for assessing functional ankle instability. However, there are some suggestions to improve the manuscript by increasing its strength and conciseness. Additionally, the manuscript contains some typographical errors.

Title:

Removing redundant words and focusing on key terms could enhance readability. For example, the words "measures" and "assessing" are somewhat repetitive. The title could be shortened without losing meaning.

Abstract:

  1. Please briefly describe the “psychometric point of view” to make it clearer.
  2. The abstract is a bit lengthy compared to the word limit suggested by the journal (<200 words); please adjust its length and make it more concise.

Introduction:

  1. Presenting the definition of ankle instability (AI), followed by its types, impacts, and so on, would help the reader easily follow the study.
  2. The term "AT" doesn't appear to be a correct term within the context. It seems to be either a typographical error or an abbreviation that wasn't explained.
  3. Adding more explanations to each questionnaire and how the interpretation of each tool would provide beneficial information for selecting the suitable one for future studies or clinical purposes.
  4. Since there are three types of ankle instability, it would be better if the authors specified the appropriate questionnaire used for each type.
  5. Please clarify why the current study focuses on the critical evaluation of the strengths and weaknesses of different measurement approaches, as there was no clear information presented in the introduction.
  6. In the last paragraph, adding the specific focus or analysis would make the objective of the study clearer.

Materials and Methods:

  1. Please explain more about the “bibliometric methods” conducted in the study.
  2. Please check the location of Figure 1, as it cuts the sentence.
  3. Enlarging the font used in Figure 1 is better since it is too small.

Results:

  1. Figure 1 should move to be part of the Materials and Methods.
  2. Adding subheadings would help the reader follow the study.
  3. In Table 1, adding columns presenting the points used to interpret or classify the ankle instability limitations and reliability of each tool would be useful.

Discussion:

  1. The paragraph is lengthy; breaking the contents according to each focused assessment or adding subheadings would help the reader follow the study.
  2. The study would benefit from a stronger narrative connecting the various sections, ensuring that the reader understands how each part contributes to addressing the research question.

Conclusion:

Using the full term of each abbreviation in this part is helpful for the audience to grasp and recall the whole piece of information.

Author Response

Dear Ms. Amy Su

RE: Measures for Assessing Functional Ankle Instability in Sport: A Critical Review and Bibliometric Analysis.

My colleagues and I would like to thank you for the opportunity to resubmit our manuscript to Healthcare. We found reviewers’ comments to be very helpful, and we have done our best to incorporate all their suggestions. We believe that this has made a significant contribution to the overall quality of the manuscript.

The reviewers’ comments and our actions are attached at the bottom of this letter. We have also included an updated version of our manuscript with all the changes highlighted in track-changes MS-word.

If you require any additional information, please do not hesitate to get in touch with us.

Thank you for considering our manuscript.

Yours sincerely,

Pedro Mendes, PhD

RESPONSE TO REVIEWER 3

Functional ankle instability (FAI) is widely researched in sports. This review examines the six most frequently cited questionnaires for assessing FAI: the Lower Extremity Functional Scale (LEFS), Foot and Ankle Ability Measure (FAAM), Foot Function Index (FFI), Foot and Ankle Outcome Score (FAOS), Olerud and Molander Ankle Score (OMAS), and Cumberland Ankle Instability Tool (CAIT). Each tool was evaluated for development, reliability, and overall summary. Bibliometric analysis showed that FAOS ranks highest and FFI lowest in the impact factor of their original publications. Despite variations in psychometric properties, all six questionnaires are robust and well-supported in the literature. Researchers should select tools based on psychometric strengths, limitations, and specific research needs, considering the impact factor as a relevant metric.

The study is of interest and provides beneficial information for those searching for questionnaires for assessing functional ankle instability. However, there are some suggestions to improve the manuscript by increasing its strength and conciseness. Additionally, the manuscript contains some typographical errors.

Authors' response: Dear reviewer, thank you for your comments and suggestions for improvement. We will do everything we can to give you the best possible response and improve the quality of our article.

Title:

Removing redundant words and focusing on key terms could enhance readability. For example, the words "measures" and "assessing" are somewhat repetitive. The title could be shortened without losing meaning.

Authors' response: Thanks for the reviewer comments. We fully agree with the reviewer comment. The title has been changed for clarity.

Action: The title has been shortened (lines 2-3): “Assessing Functional Ankle Instability in Sport: A Critical Review and Bibliometric Analysis”.

Abstract:

  1. Please briefly describe the “psychometric point of view” to make it clearer.

Authors' response:  Thank you for your comment. The questionnaires proved to be robust from a psychometric point of view in terms of its validity and reliability.

Action: To a better understanding, we have changed the sentence in lines 26-28 for: “…in terms of validity and reliability, conceptualization, structure, and usefulness, the six questionnaires proved to be robust from a psychometric point of view, being…”.

  1. The abstract is a bit lengthy compared to the word limit suggested by the journal (<200 words); please adjust its length and make it more concise.

Authors' response:  We are thankful for the reviewer’s comment. We have made some minor adjustments to the abstract in line with your comment.

Action: The abstract has been shortened to comply with the journal's guidelines, without losing the most important content. Please, see lines 17 to 30.

Introduction:

  1. Presenting the definition of ankle instability (AI), followed by its types, impacts, and so on, would help the reader easily follow the study.

Authors' response: We are thankful for the reviewer’s comment. We have improved this section of the introduction, according to the reviewers’ suggestions.

Action: We have added information to lines 39 to 42: “…is a condition characterized by a recurrent giving way of the outer (lateral) side of the ankle” and “…and can increase the risk of injuries, limitations in the regular practice of physical activity, pain, mental impact, biomechanical alterations and economic cost”.

  1. The term "AT" doesn't appear to be a correct term within the context. It seems to be either a typographical error or an abbreviation that wasn't explained.

Authors' response:  We are thankful for the reviewer’s comment. We confirm that “AT” was a typographical error.

Action: We have corrected the typographical error “AT” to “AI” (line 64).

  1. Adding more explanations to each questionnaire and how the interpretation of each tool would provide beneficial information for selecting the suitable one for future studies or clinical purposes.

Authors' response: Dear reviewer, we are grateful for your comment. However, that wasn't our aim. We just wanted to bibliometrically analyze the best questionnaire to assess FAI. Nevertheless, by looking at Table 1 and the "Development" sections in each questionnaire (3.1.1.; 3.2.1.; 3.3.1.; 3.4.1.; 3.5.1.; 3.6.1) we were able to understand some of this information.

  1. Since there are three types of ankle instability, it would be better if the authors specified the appropriate questionnaire used for each type.

Authors' response: Thanks for the reviewer comments. Unfortunately, this information is not given to us in some of the primary studies.

  1. Please clarify why the current study focuses on the critical evaluation of the strengths and weaknesses of different measurement approaches, as there was no clear information presented in the introduction.

Authors' response: Dear reviewer, thank you for your important comment. We have added information in the introduction section reinforcing the importance of assessing the strengths and weaknesses of different measurement approaches.

Action: We have added the following information in lines 74-76: “In addition, there is a wide variety of measures used to assess functional instability, which can confuse researchers and professionals in their scientific and professional activities.”

  1. In the last paragraph, adding the specific focus or analysis would make the objective of the study clearer.

Authors' response:  Dear reviewer, we have added information in the last paragraph based on your comment.

Action:  We have added the following information in lines 74-78: “In addition, there is a wide variety of measures used to assess functional instability, which can confuse researchers and professionals in their scientific and professional activities. Therefore, the purpose of this study is to identify, overview and evaluate the 6 self-report measures most frequently used to assess FAI in bibliometric terms.”

Materials and Methods:

  1. Please explain more about the “bibliometric methods” conducted in the study.

Authors' response: Thank you for your comment. In our study, we used the bibliometric methods proposed by Clancy et al. (2017). Therefore, articles published before November 2022 were searched to identify the six most highly cited FAI questionnaires. Each questionnaire was evaluated and discussed in three sections: development, reliability, and summary. Bibliometric data was obtained from the Cited Reference Search of the Web of Science. In order to evaluate the impact of the use of each questionnaire, we analyzed the total number of citations and annual citations of each questionnaire and identified the main sources of publication with impact factor, specifying the number of journals, articles, and calculating the average weighted impact factor.

Clancy, R. B., Herring, M. P., & Campbell, M. J. (2017). Motivation Measures in Sport: A Critical Review and Bibliometric Analysis. Frontiers in Psychology, 8. doi:10.3389/fpsyg.2017.00348

  1. Please check the location of Figure 1, as it cuts the sentence.

Authors' response: Thanks for the reviewer comments. The figure 1 has been replaced.

  1. Enlarging the font used in Figure 1 is better since it is too small.

Authors' response: Thank you for your comment. The figure 1 has been improved.

Results:

  1. Figure 1 should move to be part of the Materials and Methods.

Authors' response:  Dear reviewer, Figure 1 shows the results of our research. For this reason, we present it in the results section.

  1. Adding subheadings would help the reader follow the study.

Authors' response: Dear reviewer, our results section is already divided into its subtopics.

  1. In Table 1, adding columns presenting the points used to interpret or classify the ankle instability limitations and reliability of each tool would be useful.

Authors' response: Thank you. Based on the reviewer’s suggestion, we have added a column with the values from the test-retest calculation.

Action: A column labeled “Test-rretest reliability” was added to the Table 1.

Discussion:

  1. The paragraph is lengthy; breaking the contents according to each focused assessment or adding subheadings would help the reader follow the study.

Authors' response: We are thankful for the reviewer’s comment. We have improved the Discussion section, breaking the contents for clarity and adding subheadings.

Action: Please, see the tracked changes made in this section (lines 532-656).

  1. The study would benefit from a stronger narrative connecting the various sections, ensuring that the reader understands how each part contributes to addressing the research question.

Authors' response: We are thankful for the reviewer’s comment. Our study aimed to identify, overview, and evaluate the six self-report measures most frequently used to assess FAI. Thus, in the Discussion section, we start to highlight the most cited questionnaires, the main characteristics of each, and the differences between them. After that, we explored the results obtained from the bibliometric analysis of each questionnaire. We believe that the sections of the discussion allow a good understanding of the results obtained, answering the research question. This study is useful for health professionals and researchers dealing with ankle-related issues, being better prepared to identify a valid and reliable assessment tool for their clinical and research needs.

Conclusion:

Using the full term of each abbreviation in this part is helpful for the audience to grasp and recall the whole piece of information.

Authors' response:  We are thankful for the reviewer’s comment. We have reformatted the section and added the full terms.

Action: We changed “FAI” to “Functional Ankle Instability”; “FAOS” to “Foot and Ankle Outcome Score”, and “FAI” to “Functional Ankle Instability”.

Round 2

Reviewer 1 Report

Comments and Suggestions for Authors

good revision to answer my queries. 

Reviewer 2 Report

Comments and Suggestions for Authors

You have attempted to address my suggestions, but have made very few changes to the text. Therefore, I maintain my initial recommendation to reject the article.

Reviewer 3 Report

Comments and Suggestions for Authors

Thank you for addressing the concerns raised in the previous review. This revision enhances the quality and readably of the manuscript.